

**Geospatial modelling of large wood supply to rivers: a state-of-the-**
**art model comparison in Swiss mountain river catchments**
Nicolas Steeb[1,*], Virginia Ruiz-Villanueva[2,3], Alexandre Badoux[1], Christian Rickli[1], Andrea
Mini[3], Markus Stoffel[2,4,5], Dieter Rickenmann[1]
[1]Swiss Federal Research Institute WSL, Zürcherstrasse 111, CH-8903 Birmensdorf,
Switzerland
[2]C-CIA-Climate Change Impacts and Risks in the Anthropocene, Institute for Environmental
Sciences (ISE), University of Geneva, CH-1205 Geneva, Switzerland
[3]Institute of Earth Surface Dynamics (IDYST), University of Lausanne, UNIL Mouline, CH-
1015 Lausanne, Switzerland
[4]Dendrolab.ch, Department of Earth Sciences, University of Geneva, Geneva, Switzerland.
[5]Department F.-A. Forel for Environmental and Aquatic Sciences, University of Geneva,
Geneva, Switzerland
*Corresponding author: Nicolas Steeb, Email:      nicolas.steeb@wsl.ch



**ABSTRACT**

19          Different models have been used in science and practice to identify instream large wood

(LW) sources and to estimate LW supply to rivers. This contribution reviews the existing models
proposed in the last 35 years and compares two of the most recent GIS-based models by applying
them to 40 catchments in Switzerland. Both models, which we call here empirical GIS approach
(EGA) and Fuzzy-Logic GIS approach (FGA), consider landslides, debris flows, bank erosion,
and mobilization of instream wood as recruitment processes and compute volumetric estimates
of LW supply based on three different scenarios of process frequency and magnitude. Despite
being developed following similar concepts and fed with similar input data, the results from the
two models differ markedly. In general, estimated supply wood volumes were larger in each of
the scenarios when computed with the FGA and lower with the EGA models. Landslides were
the dominant process identified by the EGA, whereas bank erosion was the predominant process
according to the FGA model. These differences are discussed and results compared to available
observations coming from a unique database. Regardless of the limitations of these models, they
proved extremely useful for hazard assessment, and the design of infrastructure and other
management strategies.
**KEYWORDS:** large wood, GIS, modelling, landslide, bank erosion, debris flow, natural
hazards



## 1   INTRODUCTION

The influence of wood in watercourses is manifold. On the one hand, there are various
ecological benefits of large wood (LW), as it provides habitat and food source for many organic
organisms, thus promoting rich biodiversity (Harmon et al., 2004; Steel et al., 2003; Wondzell
and Bisson, 2003). LW also affects stream hydraulics by altering the channel morphology and
sediment control (Montgomery and Piégay, 2003; Wohl and Scott, 2016). On the other hand, large
quantities of LW may be mobilized during infrequent, high-magnitude floods and may induce
potential hazards for human settlements and infrastructure (Lucía et al., 2015c; Lucía et al., 2018;
Rickli et al., 2018; Ruiz-Villanueva et al., 2013; Steeb et al., 2017b). Consequently, river
managers are challenged to maintain a good ecological status of rivers while minimizing potential
hazards.
From a flood protection perspective, the main problem regarding LW in streams is wood
accumulation at bridges, and weirs, which reduces or even clogs the entire river cross section and
leads to backwater rise and consequent inundation (Comiti et al., 2016; Lassettre and Kondolf,
2012; Piégay et al., 1999; Rickenmann et al., 2016). The associated damage potential of LW
depends mainly on the volume of transported LW. Large wood transport is governed by the flow
conditions, river morphology (Ruiz-Villanueva et al., 2020), the size and shape of individual
wood pieces (i.e., large logs or rootwads are more prone to clogging; Bezzola et al., 2002), the
mode wood is being transported (i.e., if logs are transported congested or not; Braudrick et al.,
1997; Ruiz-Villanueva et al., 2019) and the availability or supply of wood. Wood supply occurs
by numerous geomorphic processes including bank erosion, channel migration, mass wasting
(e.g., landslides, debris flows) and natural tree mortality and fall (Benda and Sias, 2003). These
processes can be highly variable, both on temporal and spatial scales (Gasser et al., 2019).
Despite numerous existing approaches and efforts (see following section), the quantitative
estimation of LW supply volume and the definition of contributing source areas based on different
recruitment processes remain very challenging. Because LW transport happens at the end of a
long process cascade (precipitation as trigger, flood formation and recruitment processes as



supplier, channel discharge as transport medium), its estimation involves many uncertainties that
are difficult to quantify. In addition, any type of model developed to estimate and quantify wood
supply should be validated with field observations, data that is very scarce (Comiti et al., 2016;
Seo et al., 2010).
This work reviews the state-of-the-art in wood supply modelling and presents a comparison
of two recent GIS based approaches based on a similar general concept and using similar input
data. The models were validated with a unique observation dataset of supplied wood during single
events in a large number of catchments in Switzerland (Steeb, 2018; Steeb et al., 2019a). We
discuss uncertainties, limitations and strengths of the two models and compare them with other
recent approaches. In addition, we also consider implications for flood hazard assessment and
river management.

## 2   LARGE WOOD SUPPLY MODELS: A REVIEW

Over the last decades, different approaches have been developed to quantify LW supply at
both, reach and catchment scales. Gregory et al. (2003) provided a summary of the first attempts
to simulate wood supply, i.e., mostly mathematical models developed from conceptual
descriptions of selected wood recruitment processes. Later, Gasser et al. (2019) reviewed recent
approaches and evaluated whether the stabilizing effect of vegetation on total LW supply was
considered or not. Here we compile information on existing approaches and expand these
overviews to provide a review of published approaches to model recruitment processes and to
quantify LW supply (Table 1). We classify the approaches by model category (i.e., empirical,
deterministic, stochastic, or GIS-based) and summarize their main characteristics (i.e., processes
considered, spatial and temporal scales, inputs and outputs, and whether they were validated with
field observations or not). The evolution of these models illustrates and contributes to the
scientific understanding of the complex processes involved in wood supply to rivers. Some of the
earliest approaches (e.g., Malanson and Kupfer, 1993; Minor, 1997; Rainville et al., 1986; Van

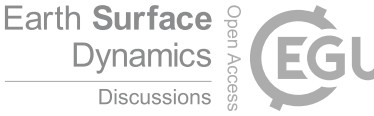

Sickle and Gregory, 1990) were designed to simulate long-term delivery of wood to river reaches
from adjacent riparian forest by tree mortality, windthrow or bank erosion. Subsequent models
attempted to describe these input processes over larger portions of river networks (Beechie et al.,
2000; Bragg, 2000; Downs and Simon, 2001; Kennard et al., 1999; Meleason et al., 2003; Welty
et al., 2002), but maintained a long-term perspective. Few studies included other processes, such
as channel avulsion (Malanson and Kupfer, 1993; Downs and Simon, 2001). These earlier models
were developed in the US, most of them in the Pacific Northwest and a few in the Southeast (e.g.,
Downs and Simon, 2001) or the Rocky Mountains (e.g., Bragg, 2000). Later, researchers started
to apply and develop models elsewhere (e.g., New Zealand; Meleason et al., 2003).
Martin and Benda (2001) and Benda and Sias (2003) were pioneers in considering mass
movements (i.e., landslides and debris flows) as wood recruitment processes, and they established
the first conceptual framework for LW budgeting. This approach has been further applied in US
mountain rivers (Benda and Bigelow, 2014; Hassan et al., 2016) before it has been adapted to
shorter timescales for mountain rivers in Italy and Switzerland (Comiti et al., 2016; Lucía et al.,
2018; Steeb et al., 2017b). Focusing on  shorter time windows and on episodic disturbances (e.g.,
floods) aggregated at the catchment scale, researchers proposed empirical equations based on
field observations of exported wood and catchment characteristics (Rickenmann, 1997; Rimböck,
2003; Steeb, 2018; Steeb et al., 2019a; Uchiogi et al., 1996). As most of the data used to derive
such empirical formulas originated from steep headwater streams and mountain rivers in
Switzerland, Austria, and Japan, application to larger catchments is associated with considerable
uncertainty.
In order to extend the analysis to larger areas, covering multiple (sub-)catchments and
applying a spatially distributed analysis, another group of models (i.e., geospatial models) used
geographic information systems (GIS) that allowed a spatially explicit assessment of different
LW recruitment processes, the identification of source areas and the estimation of LW volumes.
Rimböck (2001) developed a GIS-based model to identify potential recruitment areas of LW in
mountain streams, resulting from bank erosion, landslides and windthrow. In this approach, he



used wood volume reduction factors to distinguish between the potential LW volume (i.e.,
maximum volume that could potentially be supplied) and the estimated wood volume exported or
supplied during exceptional floods. Mazzorana et al. (2009) developed a procedure to determine
the relative propensity of mountain streams in Bolzano Province (Italy) to supply wood due to
floods, debris flows in tributaries, bank erosion and shallow landslides, based on empirical
indicators. Kasprak et al. (2012) used light detection and ranging (LiDAR) data to estimate tree
height and recruitable tree abundance throughout a watershed in US Coastal Maine, and to
determine the likelihood for the stream to recruit channel-spanning trees at the reach scale and
assess whether mass wasting or channel migration was a dominant supply mechanism. Ruiz-
Villanueva et al. (2014c) estimated potential LW volumes recruited from landslides, bank erosion
and fluvial transport during floods in the Central Mountain Range in Spain. The authors applied
a GIS model including multi-criteria and multi-objective assessments using fuzzy logic principles
together with reduction factors for predefined scenarios. The method included the analysis of the
hillslope-channel network connectivity and the resistance of the vegetation to be eroded. This
approach was recently adapted and applied to mountain catchments in Switzerland, considering
debris flows as supply processes as well (Ruiz-Villanueva and Stoffel, 2018), and it has been
further used in the present study. Also applied in Swiss mountain catchments, Steeb et al. (2017a,
2019b) proposed a GIS approach to model source areas of LW and to estimate potential supply
and exported wood volumes based on reduction factors derived from an extensive empirical
database of flood events with LW occurrence (Steeb, 2018; Steeb et al., 2019a, 2022). In
Switzerland and other countries around the Alps, some private engineering companies and
consultants, specialized on natural hazards, developed their own GIS-based models to estimate
the potential LW supply from different recruitment processes (e.g., von Glutz, 2011; Hunziker,

141 2017).

However, one important aspect of the above-mentioned GIS-based models (Mazzorana et
al., 2009; Rimböck, 2001; Ruiz-Villanueva et al., 2014c; Steeb et al., 2017a, 2019b) is that they
do not attempt to simulate the actual recruitment processes (e.g., landslides, debris flows, bank



erosion), but they used available information on areas susceptible to recruitment processes (e.g.,
from hazard maps, although these are usually derived from previous modelling studies) or expert-
based buffers. An intermediate approach was proposed by Rigon et al. (2012), who applied a
geostatistical bivariate analysis (weight of evidence method; Bonham-Carter et al., 1990) to
identify unstable areas based on weighting factors. Lucía et al. (2015a) estimated potential LW
recruitment in a mountain basin in Italy modelling shallow landslides with a hillslope stability
model (Montgomery and Dietrich, 1994) coupled to a connectivity index (Cavalli et al., 2013).
The approach was further developed by Franceschi et al. (2019) who used detailed forest
information based on a single tree extraction from LiDAR data and combined it with a 1D
hydraulic model to evaluate channel widening and LW downstream propagation. Cislaghi et al.
(2018) proposed one of the first physically-based stochastic models to simulate shallow landslides
combined with the forest stand characteristics to estimate LW recruitment from hillslopes.
Similarly, Gasser et al. (2018, 2020) proposed two frameworks to model shallow landslides, and
geotechnical and hydraulic bank erosion applying two physically-based stochastic models
together with a tree detection algorithm (Dorren, 2017) to estimate LW supply. Zischg et al.
(2018) presented a LW recruitment model coupled to a 2D hydrodynamic model to estimate LW
recruitment from bank erosion in the flood influence zone of the river. In this approach, wood
volumes were also estimated based on a single tree detection algorithm applied to a normalized
digital surface model.





**Table 1: Comparison of published wood supply models. Grey: models used for comparison in**
**this work**

| Reference | Country | Model name | Category | Processes considered | Spatial scale | Temporal scale | Main input variables | Output |
|---|---|---|---|---|---|---|---|---|
| Rainville et al., 1986 | USA (Pacific Northwest) | Not specified | Deterministic | Tree fall | Stream reach | between 25 and 300 years (time steps of 10 years) | Not specified | Number of wood pieces |
| Murphy and Koski, 1989 | SE Alaska | Not specified | Deterministic | Tree fall and bank erosion | Stream reach | 250 years (time steps 1 year) | Survey measurements; channel width, wood diameter, forest stand | Number of wood pieces |
| Van Sickle and Gregory, 1990 | USA (OR) | Not specified | Stochastic | Tree fall | Stream reach | time steps of 10 years | Riparian stand density, tree height, stream length | Number of wood pieces |
| Malanson and Kupfer, 1993 | USA | FORFLO model | Stochastic | Tree fall | Stream reach / floodplain | 500 years (time steps 1 year) | Tree species, tree height, diameter, water level | Biomass |
| Rickenmann, 1997 | Switzerland, Japan, USA | Not specified | Empirical | Wood export, (recruitment process not specified) | Catchment | Event | Catchment area, forested catchment area, stream length, forested stream length, peakflow, flood runoff and bedload volume | LW potential (instream wood), estimated LW supply volumes |
| Beechie et al., 2000 (based on Kennard et al., 1999) | USA (WA) | Riparian-in-a-Box | Deterministic | Natural tree mortality, windthrow, bank erosion | Stream reach | 150 years (time steps 10 years) | Tree species, diameter, height, and crown ratio in stands; site/reach geometry | Number of wood pieces and LW volume |
| Bragg, 2000 | USA (Inter-mountain West) | CWD model (1.2) | Stochastic | Episodic tree mortality (spruce beetle outbreak, moderately intense fire, and clear-cut) | Stream reach | 300 years (time steps 10 years) | Stand density, species, tree height & diameter | Number of wood pieces and LW volume |
| Downs and Simon, 2001 (based on earlier models of Simon, 1989 and Hupp and Simon, 1991) | USA (MS) | Simon channel evolution model | Deterministic | Bank erosion and channel avulsion | Stream reach / river network | time steps of 10 years | Channel morphology surveys, rates of knickpoint migration, quantitative characteristics of riparian vegetation | Number of wood pieces and LW volume |
| Rimböck, 2001 | Germany (Bavarian Alps) | *Luftbildbasierte Abschätzung des Schwemmholzpotenzials* | Deterministic GIS-based | Bank erosion, mass | Stream reach | Event | DTM, stand density | LW potential volume |





| | | | | failures (i.e., landslides), windthrow, avalanches | | | | |
|---|---|---|---|---|---|---|---|---|
| Welty et al., 2002 (same model as Kennard et al., 1999 and Beechie et al., 2000) | USA (Pacific Northwest) | Riparian aquatic interaction simulator RAIS | Deterministic | Natural tree mortality, windthrow, bank erosion, mass failures | Stream reach | 240 years (time steps 10 years) | Various variables describing forest stand, stream width, initial LW, conifer/hardwood depletion rate, zone widths, windthrow rate, fall direction bias, LW placement option | Number of wood pieces and LW volume |
| Benda and Sias, 2003 | USA (Pacific Northwest) | Not specified | Deterministic | Episodic tree mortality (e.g., fire, wind), bank erosion, mass failures and debris flows | Catchment and stream reach | 800-1800 years (time steps 10 years) | Stand density, tree height, channel width, recruitment area & rates | Number of wood pieces and LW volume |
| Meleason et al., 2003 | USA (Pacific Northwest) | Streamwood | Stochastic | Tree fall by natural mortality | Stream reach | 500 years (time steps 10 years) | List of trees that died in a year (wood model input = forest model output) | Number of wood pieces and LW volume |
| Benda et al., 2007 | USA | NetMap | Deterministic GIS-based | Hillslope erosion, sediment and wood supply | Watershed, subbasin, or stream segment | Not specified | Base terrain parameters including DEM and climate data | LW accumulation type |
| Mazzorana et al., 2009 | Italy (Autonomous Province of Bolzano) | Not specified | Deterministic GIS-based | Bank erosion, mass failures and debris flows | Catchment | Event | DTM, hazard index map (debris flow, overbank sedimentation), land use map, stand map, torrent network map | Hazard index maps classifying torrent catchments according to propensity to entrain and deliver woody material. |
| Eaton et al., 2012 | British Columbia | The reach-scale channel simulator (RSCS) was | Stochastic (Monte Carlo) | Tree fall by natural mortality | Stream reach | One year time step | Tree height, tree diameter, tree fall orientation, forest density, chronic mortality, decay and breakage | Wood load (m³·m⁻²) and jam formation |
| Kasprak et al., 2012 | USA (ME) | Not specified | Deterministic | Bank erosion, mass failures and debris flows | Both stream reach and catchment | 100 years period | Stand data, LiDAR DEM | Number of wood pieces |
| Rigon et al., 2012 | Italy (Eastern Alps) | Not specified | Geostatistical, GIS-based | Mass failures (i.e., landslides) | Both stream reach and catchment | Event | Landslide and debris flow inventory data, stand data, DEM | LW volume |



| | | | | | | | | |
|---|---|---|---|---|---|---|---|---|
| Benda and Bigelow, 2014 (same model as Benda and Sias, 2003) | USA (CA) | Not specified | Deterministic | Tree mortality, bank erosion, mass failures, debris flows and snow avalanches | Both stream reach and catchment | 100 years period | Survey measurements | Quantification of wood recruitment, storage and transport |
| Ruiz-Villanueva et al., 2014c | Spain | Not specified | Fuzzy-logic GIS-based | Fluvial transport, bank erosion and mass failures (i.e., landslides) | Catchment | Event | DEM, topography, natural hazards maps, geomorphological units, forest density, tree species, height & diameter | Number of wood pieces, LW volume |
| Lucía et al., 2015a | Italy (North-western Ap-ennines) | Not specified | Deterministic GIS-based | Bank erosion, mass failures | Catchment | Event | DTM, DSM (digital surface model) | LW volume |
| Benda et al., 2016 (sensu Benda and Sias 2003) | USA (OR) | Reach Scale Wood Model (RSWM) | Deterministic | Tree fall by natural mortality | Stream reach | 100 years (5-year time steps) | Stand density, mortality rate, tree height & diameter, slope, stream width | Instream wood quantity (pieces and volume) |
| Hassan et al., 2016 (budget concept used in Benda and Sias 2003) | Canada (BC) | Not specified | Deterministic | Tree mortality, bank erosion, mass failures | Stream reach | 100 years period | High field data requirements, most can be obtained from air photo measure-ments, forest inventory data, and/or regional values | LW volume |
| Steeb et al., 2017b; Steeb, 2018 (updated and expanded from Rickenmann, 1997) | Switzerland, Italy, France, Germany Japan | Not specified | Deterministic, Empirical | Wood export, (recruitment process not specified) | Catchment | Event | Catchment characteristics, flood event characteristics | LW volume |
| Steeb et al., 2017a, 2019b | Switzerland (Alps) | Empirical GIS Approach (EGA) | Deterministic GIS-based, empirical | Bank erosion, mass failures, debris flows | Catchment | Event | SilvaProtect-CH data, stream network, catchment area, ecomorphology data, stand data (NFI) | LW volume, number of wood pieces, recruitment areas, proportion of fresh and dead wood |
| Cislaghi et al., 2018 | Italy (Eastern Alps) | Combination of the probabilistic multidimensional stability model PRIMULA and a hillslope-channel transfer mode | Stochastic | Mass failures (i.e., landslides) | Catchment | Event | DEM, geological map, rainfall, forest stand characteristics | LW volume |
| Gasser et al., 2018 and 2020 | Switzerland | SlideforMAP, BankforMAP, FINT | Stochastic | Bank erosion, | Stream reach | Event | DTM, DSM, precipitation maps, soil map, | LW volume |





| | | | | mass failures | | | vegetation efficiency (erosion prevention) | |
|---|---|---|---|---|---|---|---|---|
| Ruiz-Villanueva and Stoffel, 2018 | Switzerland | Fuzzy-Logic large wood recruitment Toolbox (here Fuzzy-logic GIS Approach; FGA) | Fuzzy-logic GIS-based, | Bank erosion, mass failures, debris flows | Catchment | Event | SilvaProtect-CH data, stream network, catchment area, DEM, ecomorphology data, stand data (NFI) | LW volume, recruitment areas |
| Zischg et al., 2018 | Switzerland | LWDsimR (coupled with Basement-ETH) | Deterministic | Bank erosion | Stream reach | Event | DEM, hydrograph, forest stand | LW volume |
| Franceschi et al., 2019 (based on the model developed by Lucía et al. 2015) | Italy (South Tyrol) | Not specified | Deterministic GIS-based | Bank erosion, mass failures | Catchment | Event | DTM, geomorphological map, precipitation, discharge | LW volume |




## 3 GEOSPATIAL MODELLING OF LARGE WOOD SUPPLY IN SWISS MOUNTAIN CATCHMENTS

### 3.1 General concept

In this contribution, two LW models were compared; the empirical GIS approach (EGA) by Steeb et al. (2017a, 2019b) and the Fuzzy-Logic GIS approach (FGA) by Ruiz-Villanueva and Stoffel (2018) which is a variation of the model presented by Ruiz-Villanueva et al. (2014c). Both, the EGA and FGA are based on a similar general concept (Figure 1) and fed with similar input data and defined equivalent scenarios (see following subsections) to make comparison possible. Both models were developed in the context of *WoodFlow*, a Swiss research program aimed at creating knowledge and methods to analyze instream wood dynamics, with particular attention to watercourses in the Alpine region (FOEN, 2019).

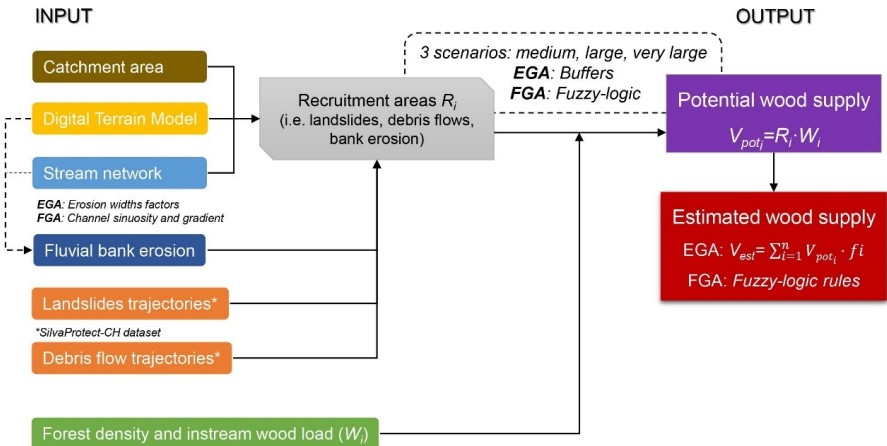

**Figure 1: Conceptual model of the empirical GIS approach (EGA) and the Fuzzy-Logic GIS approach (FGA).** $V_{pot}$ = potential wood supply [m³]; $V_{est}$ = estimated supplied wood [m³]; $i$ = recruitment process[-]; $R$ = recruitment area [ha]; $W$ = forest density or instream wood load [m³ ha⁻¹]; $f$ = volume reduction factor [-].

The general concepts and main steps of the GIS-based approaches were to (i) identify the recruitment areas on the hillslopes and along the stream network that may contribute woody material to streams, such as areas affected by landslides, debris flows and bank erosion; (ii) create



three different scenarios based on the process frequency and magnitude; and to (iii) provide
estimates of potential LW supply $V_{pot}$ (i.e., worst case scenarios) and supplied wood volumes for
each scenario $V_{est}$. The methods aim at estimating supply wood volumes at the catchment scale
and do not include the analysis of wood transfer (i.e., transport and deposition) through the stream
network.
Potential large wood supply $V_{pot}$ was calculated by intersection of the modelled recruitment
areas with forest cover. During a flood, however, only a part of the LW potential is actually
recruited and exported out of the catchment. Therefore, empirically derived volume reduction
factors (EGA) or fuzzy logic principles (FGA) were applied to best estimate actual supplied LW
volumes $V_{est}$. Modelling results were validated by comparison with available empirical data
documented after flood events (Table S1 in supplementary material).

**3.2 Input data**
*3.2.1 Catchment areas and stream network*
The topographical catchment areas (feature polygons), which define the perimeters of
investigation, were available from the geodataset "topographical catchments of Swiss
waterbodies" (FOEN, 2015). The stream network of Switzerland at a scale of 1:25,000
(swissTLM3D, ⓒ 2016 swisstopo [DV033594]) was pre-processed by adding information on
channel width as derived from a Swiss-wide ecomorphological dataset (Ökomorphologie
Stufe F ©FOEN; Zeh Weissmann et al., 2009). Based on this dataset, the channel width was
known for 42 % (25,800 km) of the total Swiss streams' length. For the remaining 58 %, we
extrapolated channel width based on stream order (Strahler, 1957) and altitude classes (Table S2).
The stream network and channel widths were used to define intersections and connectivity
between the hillslopes processes and the streams, to estimate the bank erosion prone areas
(sections 3.3 and 3.4) and to assign values of instream dead wood volumes (section 3.2.3).



*3.2.2  SilvaProtect-CH and the identification of landslide and debris flow*
*trajectories*

For the modelling of the two recruitment process categories landslide and debris flow, both

GIS models used the SilvaProtect-CH dataset from Losey and Wehrli (2013). As part of the
SilvaProtect-CH project, several natural hazard processes were modelled over the entire Swiss
territory using partly physically-based models. As a result, process trajectories that describe the
topographic flow path and runout distances (from starting to deposition zone) of the investigated
natural hazard processes were readily available (details are provided in the supplementary
material). These trajectories were processed further to identify potential recruitment areas of LW
supply (sections 3.3. and 3.4).

*3.2.3  Forest density and instream wood load*

The density of living trees in Swiss forests [$m^3\ ha^{-1}$] was derived from a Swiss nationwide

raster map with an original resolution of 25 x 25 m (rescaled to 1 x 1 m; Figure 2). The raster map
is based on a growing stock model developed by Ginzler et al. (2019) that quantifies forest density
in relation to tree height (based on airborne stereo imagery), canopy cover, topographic position
index, mean summer temperature and elevation. The EGA and FGA models further consider an
estimate of deadwood on the forest floor [$m^3\ ha^{-1}$] (i.e., equal to 5% of living trees density) based
on empirical data of the Swiss National Forest Inventory (NFI; WSL, 2016).

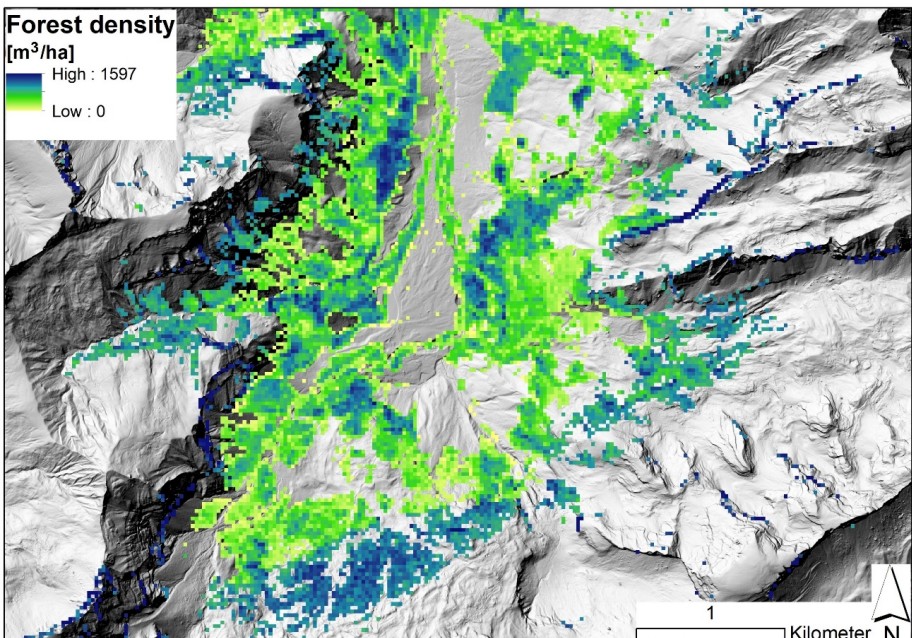


**Figure 2: Snapshot of the wood stock raster map in the Grosse Melchaa catchment near**
**Stöckalp (Canton Obwalden). Background: Digital terrain model (hillshade), © swisstopo.**

Additionally, instream wood loads were included in the calculations, accounting for

potential LW volumes from accumulated deadwood in the channel. Detailed information on wood
loads across the stream network was not available, so based on a literature review by Rickli and
Bucher (2006) and Ruiz-Villanueva et al. (2016), volumes of instream wood were assigned to the
different streams grouped by channel width (EGA) or by stream order (FGA) classes (see
following sections).

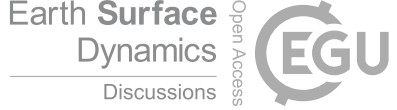

### 3.3 The empirical GIS approach (EGA)

Debris flow and landslide trajectories from SilvaProtect-CH were constrained by intersection with the stream network and forest cover. Only landslide trajectories with starting points within a 50-m distance from the stream network were considered. This limitation was supported by the landslide database of Rickli et al. (2016) where 44 % of all documented landslides showed a runout distance of less than 50 m (around 80 % are within a distance of 100 m). For each scenario (section 3.5), different buffer widths $w_b$ were applied on both sides of the relevant debris flow and landslide trajectories (i.e., medium scenario: $w_b$ = 5 m; large scenario: $w_b$ = 10 m; very large scenario: $w_b$ = 15 m). The buffer widths were chosen in ranges according to the Swiss landslide database (Rickli et al., 2016). Potential recruitment areas were finally extracted as the overlap of the buffered trajectories with the forest layer.

The extent of bank erosion in EGA was assumed to be proportional to the given channel width. Scenario-specific erosion width factors $e_w$ (i.e., a multiple of the channel width) were empirically derived from observations after the well-documented August 2005 flood in Switzerland, for which a large dataset was analysed and made available (Bachmann Walker, 2012; Hunzinger and Durrer, 2008). Scenario-specific erosion width factors were $e_w$ = 1.5 for the medium scenario, $e_w$ = 3 for the large scenario, and $e_w$ = 4.5 for the very large scenario. The resulting buffer widths were added to the original channel width. Potential recruitment areas due to bank erosion were finally extracted as the overlap of the buffered stream network with the forest layer.

The estimation of previously stored wood load within the river network (i.e., instream deadwood) was based on empirical values of wood storage per stream hectare. Rickli et al. (2018) documented instream wood storage for ten reaches in Swiss torrents. This database was complemented with 39 additional values from various other European rivers, based on a literature review by Ruiz-Villanueva et al. (2016), in order to have reliable derivations. Finally, we assigned wood load values into three channel width classes (i.e., <5 m = 94 $m^3 \cdot ha^{-1}$; 5-10 m = 67 $m^3 \cdot ha^{-1}$; >10 m = 42 $m^3 \cdot ha^{-1}$).



Potential source areas from different recruitment processes may partly overlap. For this
reason, a priority sequence was determined so that such overlapping areas were not counted more
than once. This was defined according to the following principle: The closer to the channel a
recruitment process occurs, the higher the priority: instream wood > debris flow > bank erosion
> landslide. For example, overlapping areas of debris flows and bank erosion were assigned to
the process area debris flow.
Potential recruitment areas were finally used to calculate the potential LW supply $V_{pot}$ by
multiplying the process areas with the respective forest density (for debris flows, landslides and
bank erosion) or wood load (for instream deadwood). From the resulting potential LW supply,
the actual LW supply $V_{est}$ was estimated. To do so, volume reduction factors $f$ were used, which
assumed different values depending on the recruitment process and scenario of process magnitude
(Table 2). The volume reduction factors were empirically determined with three different
approaches (Steeb et al., 2019b): 1) Comparison with literature data, including values from other
studies and models that proposed reduction factors; 2) comparison of potential vs. observed
recruitment areas; and 3) comparison of estimated vs. observed wood volumes from previous
floods (see the five blue catchments in Figure 3).
Values of observed LW supply volumes and recruitment areas together with the associated
catchment and flood specific parameters were taken from a complementary empirical dataset that
was also part of the WoodFlow research program. In total, the LW database consisted of 210 data
entries. Most entries (171) refer to events in Switzerland. Also included are flood events from
Japan, Italy, Germany and France (Steeb et al., 2019a).



**Table 2: Overview of volume reduction factors *f*, classified by scenario and recruitment**
**processes.**

| Scenario | Instream wood | Debris flow | Bank erosion | Landslide |
|---|---|---|---|---|
| **Medium** | 0.10 | 0.05 | 0.05 | 0.01 |
| **Large** | 0.30 | 0.10 | 0.10 | 0.05 |
| **Very large** | 0.70 | 0.30 | 0.20 | 0.10 |


The EGA model has been originally developed with ArcGIS 10.1 (©ESRI) and updated
with ArcGIS 10.8 (©ESRI). The toolbox is freely available for download on the website
*www.woodflow.ch*.

**3.4    The Fuzzy-Logic GIS approach (FGA)**
The areas prone to landslides and debris flows were defined based on the linear trajectories
provided by the SilvaProtect-CH database. To transform these lines into areas (i.e., pixels, as the
FGA is entirely raster based), the density of the lines was used to classify the terrain into three
intensity scenarios (section 3.5). High trajectory density was assumed to represent areas that are
more prone to landslides or debris flows, more likely of a higher frequency and therefore, lower
magnitude. Low trajectory density was assumed to represent areas that are less prone to mass
movements, more likely affected by higher magnitude and thus lower frequency events. The
thresholds to classify the three areas was based on four natural breaks (Figure S1A in
supplementary material). In the case of mass movements, the delivery of wood to the stream
network depends not only on the area of the landslide, but also on its connectivity to the channel
(Ruiz-Villanueva et al., 2014c). Once the trajectories were converted to density pixels, the
connectivity between these pixels and the stream network was established for landslide-prone
pixels, as a function of both the distance to the channel and the terrain slope. In addition, a buffer
area of influence was also established around these areas, to include toppled trees that may be



recruited indirectly by the action of landslides. Trees located in a landslide-prone pixel or in the
toppling influence area (defined as a buffer equal two times the mean tree high), may reach the
channel if they were close enough (Euclidean distance < 50 m) or further away but on a steep
slope (>40%;). In the case of debris flows, all pixels were assumed to be connected to the stream
network.
Areas prone to bank erosion were computed based on channel sinuosity and gradient (as
proxies for channel lateral migration and transport capacity; Ruiz-Villanueva et al., 2014c), the
channel width and a defined width ratio. The width ratio was used to estimate the potential
resulting channel width after bank erosion during floods. It was calculated analysing an European
database (Ruiz-Villanueva et al., *in prep.*), including several rivers and flood events in
Switzerland and other 6 countries, and three scenarios were defined for different channel width
classes (9 classes ranging from < 3 to > 50 m). The stream network provided by the
ecomorphology database (section 3.2.1) was grouped by the channel width classes considered and
the width ratio was assigned to estimate the resulting potential erodible width for each stream
segment (Figure S1). The width ratio (ranging between 1 and 4) generally increases with scenario
intensity and decreasing channel width. The resulting buffers were transformed to pixels and the
final pixels prone to bank erosion were assigned based on channel sinuosity and gradient. Stream
segments characterized with high sinuosity and high gradient were assumed to be more prone to
bank erosion.
The described variables (i.e., landslide prone areas, connectivity, debris flow prone areas,
bank erosion prone areas, sinuosity and gradient) were transformed to fuzzy sets using the Fuzzy
Membership tool initially developed in ArcGIS 10.1 and updated to ArcGIS 10.7 (©ESRI) with
a linear membership function. The resulting converted fuzzy variables were combined (e.g.,
landslides prone pixels and connectivity; sinuosity and gradient) with the Fuzzy Overlay tool
(©ESRI). As a result, all pixels were transformed to fuzzy values ranging from 0 to 1; they were
then used to compute the volume of wood by multiplying the fuzzy pixel value by the forest
density pixel value (section 3.2.3). In case of overlaping pixels, priority was given to areas prone



to debris flows, then bank erosion and finally landslides (as in the EGA approach). The final
calculation considered also the accumulated wood load within the river network, but applying a
slightly dfferent approach than for the EGA. This was estimated by assigning wood load values
based on literature (Ruiz-Villanueva et al., 2016) to the different river segments grouped by
stream order classes (i.e., < 3 stream order: 60 $m^3 \cdot ha^{-1}$; between 3 and 6 order: 120 $m^3 \cdot ha^{-1}$; > 6
order: 50 $m^3 \cdot ha^{-1}$) and multiplied by fuzzy layers.
**3.5    Model scenarios definition**
Three different scenarios were designed to estimate supplied wood volumes, based on a
qualitative assessment of the frequency and intensity of the wood recruitment processes involved.
These scenarios are called: medium scenario (medium-to-high frequency and intermediate
magnitude), large scenario (relatively low frequency and medium-to-high magnitude), and very
large scenario (very low frequency and very high magnitude)
Most of the documented floods with LW occurrence that were used to validate the GIS
models had a precipitation and/or peak runoff return period of 50-150 years, which was assigned
to the large volume scenario. The other two scenarios refer to approximate return periods and
were determined using *ad hoc* volume reduction factors (EGA) or the fuzzy logic rules (FGA),
because they could not be quantified more precisely due to a lack of data.
In addition to the estimated supplied wood volumes for each scenario, a potential wood
volume was also computed. The potential volume was assumed to be the maximum wood volume
supplied at the catchment scale, computed without any reduction by a coefficient (EGA) or by the
fuzzy logic values (FGA).
**3.6    Test catchments**
In the 40 catchments analysed in this work (Figure 3), considerable amounts of LW were
recruited and transported during past floods, and the resulting LW volumes were well documented
(mainly from the August 2005 flood; Rickli et al. 2018 and Steeb et al., 2017b). Table S1 in the
supplementary material provides an overview of the 40 test catchments and their characteristics.



Earth **Surface**
Dynamics
Discussions

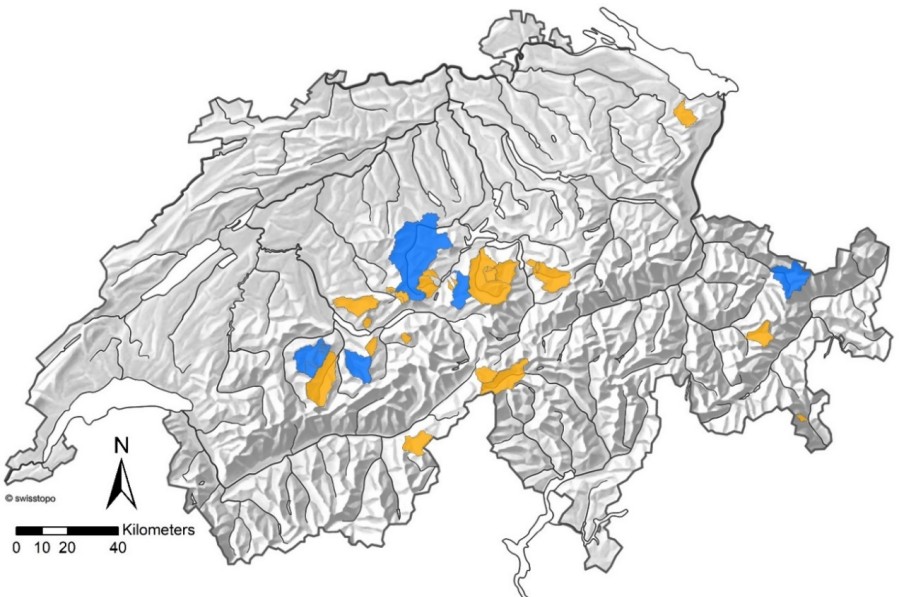


**Figure 3: Location of the 40 test catchments (orange; with many nested sub-catchments). The five catchments in blue (Chiene, Chirel, Grosse Melchaa, Landquart, Kleine Emme) were used to calibrate the volume reduction factors from the EGA approach so that the estimated supplied wood was in the same order of magnitude as the observed values from past flood events.**

**3.7. Model results analysis**

Model results were first compared to observed wood volumes during floods, and then analysed in terms of (modelled) wood volumes per scenario, potential wood volume, wood volume supplied by different recruitment or supply processes (i.e., landslides, debris flows and bank erosion), and the estimated instream wood volume.

Statistical analyses were realized with the software RStudioVersion 2021.9.0.351 (R Studio Team, 2021). Differences between the two models and between them and the available observations were tested by the nonparametric Wilcox (Mann-Whitney) or Kruskal-Wallis tests for two or more groups respectively (Stats package; R Core Team, 2019). Significance was set to a p value <0.05. The dependence of wood volume on catchment controlling variables was verified



by means of scatter plots, regression analysis and correlation (*ggally* package; Schloerke et al.,

2021).


## 4    RESULTS

### 4.1    Comparison between model outputs and model approaches (EGA/FGA)

The two GIS approaches provide geospatial outputs – EGA in the form of feature class
polygons and the FGA in pixel-based raster files – that can be visualized on a map, as shown in
Figure 4. Potential recruitment areas for debris flow, landslide and bank erosion are generally
larger for EGA, i.e., the defined EGA buffer widths provide more supply-prone areas than the
respective combination of FGA fuzzy layers within the same perimeter.

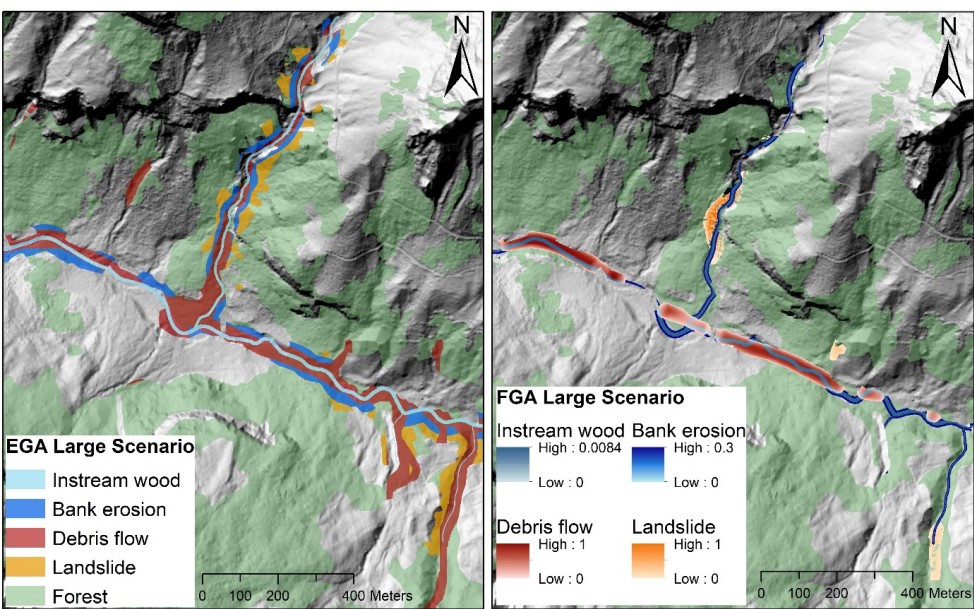


**Figure 4: Large volume scenario comparison of model outputs from EGA (left) and FGA (right)**
**at the Spiggebach torrent within the Chiene river catchment (Canton Bern). Potential recruitment**
**areas are shown for landslides (orange), debris flows (red), and bank erosion (dark blue). The stream**
**network (light blue) includes also instream wood loads. Background: Digital terrain model**
**(hillshade), © swisstopo.**





The estimated supply and potential wood volumes for the three scenarios and the two models
are shown in Figure 5 together with the available observations. The comparison between modelled
and observed wood volumes is presented in section 4.3, the focus here is on differences between
the two models. In general terms, Figure 5a highlights that the estimated supply wood volumes
for each scenario were larger when computed by the FGA and lower by the EGA. For example,
for the medium scenario, the averaged wood volumes were 994 m$^3$ and 3318 m$^3$ for EGA and
FGA, respectively. The differences were slightly reduced for the other two scenarios, for which
volumes equal to 7127 m$^3$, 17353 m$^3$, 8199 m$^3$ and 19712 m$^3$ were obtained (for the large and
very large scenarios and the EGA and FGA, respectively; Table 3).

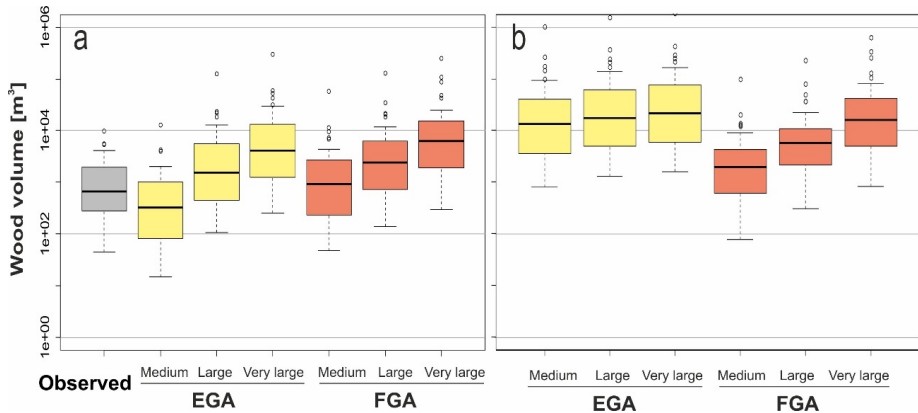


**Figure 5: Boxplots of wood supply (a) and potential (b) volume (m$^3$) estimated by the two models**
**EGA and FGA, and the three scenarios (i.e., medium, large, very large). Observed LW supply during**
**past events for all studied catchments (n=40) is given in grey color in panel (a).**





**Table 3: Observed and estimated LW supply volumes for the three scenarios (i.e., medium,**
**large, and very large) and the two models (i.e., EGA and FGA) for all studied catchments.**

| Wood supply volume [m³] | Observed | EGA | | | FGA | | |
|---|---|---|---|---|---|---|---|
| | | Medium | Large | Very large | Medium | Large | Very large |
| Min. | 45 | 15 | 106 | 253 | 48 | 141 | 300 |
| 1st | 290 | 83 | 475 | 1378 | 244 | 764 | 2037 |
| Median | 673 | 329 | 1562 | 4189 | 921 | 2430 | 6342 |
| Mean | 1428 | 994 | 7127 | 17353 | 3318 | 8199 | 19712 |
| 3rd | 1906 | 967 | 5161 | 12609 | 2588 | 6083 | 15191 |
| Max. | 9741 | 12757 | 126648 | 296893 | 57152 | 128575 | 249256 |


Significantly higher values were computed for the large and very large scenarios compared
to the medium scenarios, with a similar pattern shown by the two models. Larger differences were
observed when comparing the estimated potential volumes (Figure 5b and Table 4). In this case
the EGA resulted in much higher values than the FGA (especially for medium and large
scenarios), which is a result of much larger potential recruitment areas (Figure 4). Figure S3 shows
that for EGA, the estimated LW supply volume corresponds to 8 % of the potential wood supply
volume on average. In the case of FGA, this ratio varies much more with an average of 47 %.

**Table 4: Potential LW supply volumes for the three scenarios (i.e., medium, large, and very**
**large) and the two models (i.e., EGA and FGA) for all studied catchments.**

| Potential wood volume [m³] | EGA | | | FGA | | |
|---|---|---|---|---|---|---|
| | Medium | Large | Very large | Medium | Large | Very large |
| Min. | 807 | 1289 | 1601 | 76 | 305 | 811 |
| 1st | 3529 | 4949 | 6000 | 613 | 2203 | 5341 |
| Median | 13226 | 17579 | 21619 | 1965 | 5774 | 15965 |
| Mean | 58664 | 86984 | 105723 | 5961 | 16173 | 52995 |
| 3rd | 37672 | 59612 | 74948 | 4207 | 10665 | 41066 |
| Max. | 1011306 | 1534850 | 1866295 | 100165 | 231336 | 632151 |





### 4.2 Contribution from different supply processes


The main difference between the two models was the estimated contribution from each
supply process to the obtained wood volume. Landslides were the dominant process in the case
of the EGA, with a contribution up to more than 60% of the computed wood volume (for the large
scenario); whereas bank erosion was the predominant process in the FGA model for all scenarios
(Figure 6). Debris flows played an intermediate role in supplying wood according to the two
models; however, the importance of this process varied depending on the scenario. For the
medium scenario, the EGA model showed a similar percentage of averaged wood supplied by
landslides and debris flows. The FGA, contrastingly, computed most of the averaged wood
volume supplied by bank erosion, and only a low percentage of wood supplied by landslides and
debris flows. Only for the very large scenario, the importance of landslides, in terms of percentage
of supplied wood, equaled or even exceeded, the volume estimated from bank erosion with the
FGA.

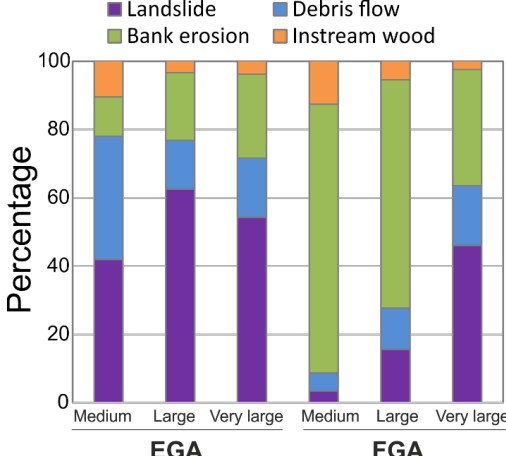


**Figure 6: Large wood volumes supplied by each process, model, and scenario averaged for all 40 study sites.**



The difference between the contribution of each process to the estimated volumes is clearly
shown in Figure 7 and 8 (with FGA resulting in generally higher volumes than EGA). The graph
illustrates that statistically significant differences were found between the computed supply wood



volumes by the two models and by bank erosion process. The median wood supply values (see
black lines within boxplots of Figure 7) are about a factor of 1000 and 10 larger for the FGA than
for the EGA, and for the medium and large scenarios respectively. This explains the relative
dominance of bank erosion for the FGA (see also Figure 8), for the medium and large scenario.
The wood volumes supplied by the other processes were not significantly different between the
two models. Only the estimated instream wood volume for the medium scenario showed a
significant difference between the EGA and the FGA, with larger volumes computed by the latter.

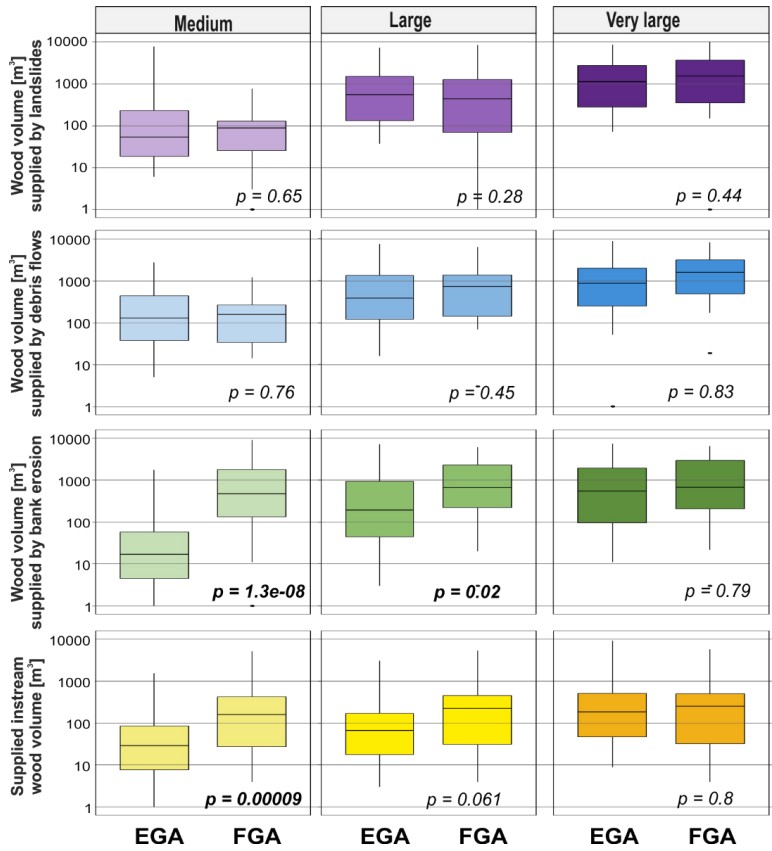

**Figure 7: Wood volumes supply estimated for landslides, debris flows, bank erosion, and**
**estimated supplied instream wood by the two models and the three scenarios. The p-value is from the**
**Wilcoxon test (significant values shown in bold).**



However, the contribution of each process to the computed wood volume did not only vary
according to the model, but also according to the site. Figure 8 shows a selected sub-dataset of
catchments with different drainage areas, revealing the large variability of the dominant wood
supply process, and the dominance of different processes over the others in the two models. In
general, the FGA approach shows a larger contribution from landslides and debris flows in smaller
catchments, while landslides are the major contributor to wood supply regardless the catchment
size for the EGA. Bank erosion is a minor contributor to the estimated supply in EGA for most
sites and irrespective of the scenario used. However, bank erosion is the most relevant process for
the FGA, which is clearly illustrated by the Kleine Emme River catchment, the largest of the study
sites of the dataset, for which the FGA estimates the largest contribution by this process. The
EGA model, on the other hand, estimated a larger contribution from landslides for this site.
The proportion of instream wood loads remains constant, independent of catchment size (2-
13 % of total wood supply). The contribution of debris flows and landslides are highly variable
depending on topography, and can be dominant for small (e.g., Secklisbach) or large catchments
(e.g., Grosse Melchaa or Chirel).

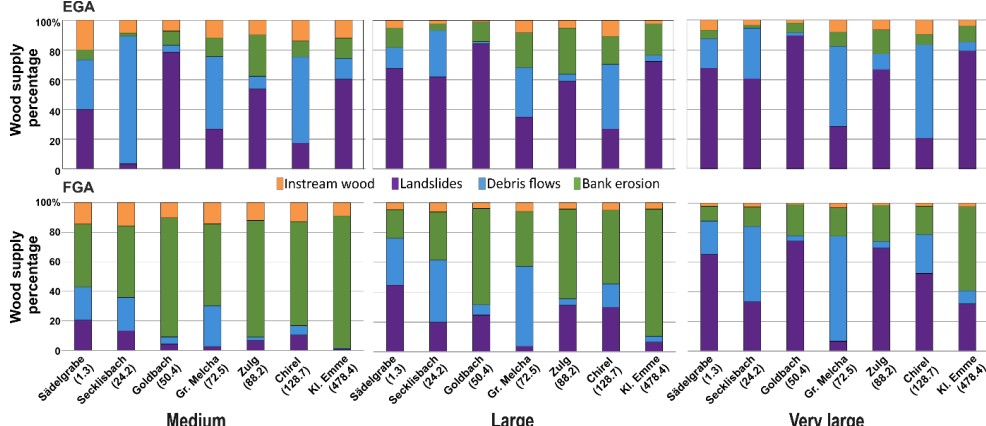


**Figure 8: Percentage of wood volume supplied by each process, model, and scenario for selected**
**studied sites, the names and catchment area in (km²) are provided in the abscissa.**

### 4.3    Estimated and observed wood volumes

The comparison between observed LW volumes $V_{obs}$ and estimated (modelled) LW volumes
$V_{est}$ are shown in Figure 9a. There is a relatively large scattering when comparing observed and
estimated wood loads. Both under- and overestimation of $V_{obs}$ are observed for both models, with
a larger tendency for overestimation. Overestimation remains generally within two orders of



486 magnitude (typically higher values for FGA), underestimation within one order of magnitude

487 (typically lower values for EGA).

488  Figure 9b further shows the ratio of $V_{est}/V_{obs}$ versus catchment area. Both under- and

489 overestimation of $V_{obs}$ are present over >2 order of magnitude for all catchment areas. However,

490 in general, overestimation increases with increasing catchment size for both models. There is a

491 shift around a catchment area of 7 km$^2$, above which overestimation is significantly larger (with

492 a factor of >10). In catchments with areas less than 7 km$^2$, estimated wood supply is generally

493 underestimated (see dashed line in Figure 9b).

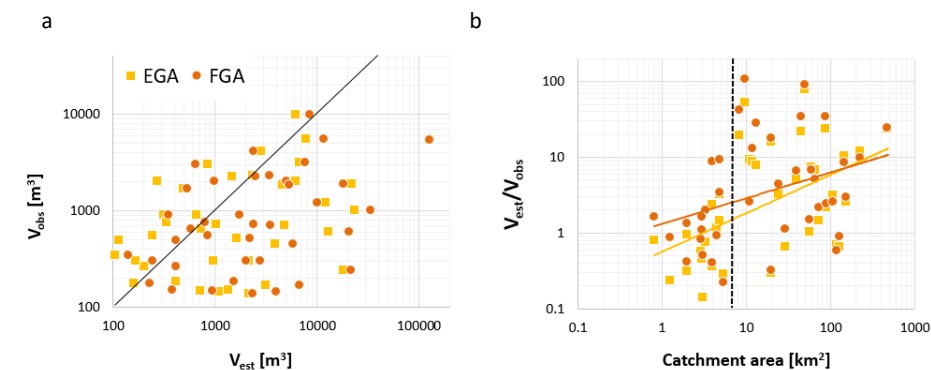

496 **Figure 9: Left: Modelled LW $V_{est}$ (large scenario) versus observed wood volume $V_{obs}$ during past**
497 **events. The black line shows the line of equality (1:1 line). Right: Ratio of $V_{est}/V_{obs}$ versus catchment**
498 **area.**

500  This tendency of overestimation with increasing catchment size can also be explained by

501 comparing the ratio of observed and potential wood volume $V_{obs}/V_{pot}$ versus catchment area

502 (Figure 10a). With increasing catchment size, there is a trend of decreasing ratio values of

503 $V_{obs}/V_{pot}$. This means in larger catchments, the volume reduction factors (FGA) and the fuzzy rules

504 (FGA) are often not small enough to reduce the wood potential accordingly, creating

505 overestimation of wood volumes ($V_{est} > V_{obs}$).

506  Since potential wood volumes are much higher for EGA (Table 4 & Figure 5b), the ratio of

507 $V_{obs}/V_{pot}$ is also much smaller in case of EGA (almost one order of magnitude difference as shown





in Figure 10b). For FGA few examples (i.e., six orange dots in Figure 10a) exist for which the
potential wood volume is even smaller than the observed wood volume ($V_{obs}/V_{pot} > 1$).

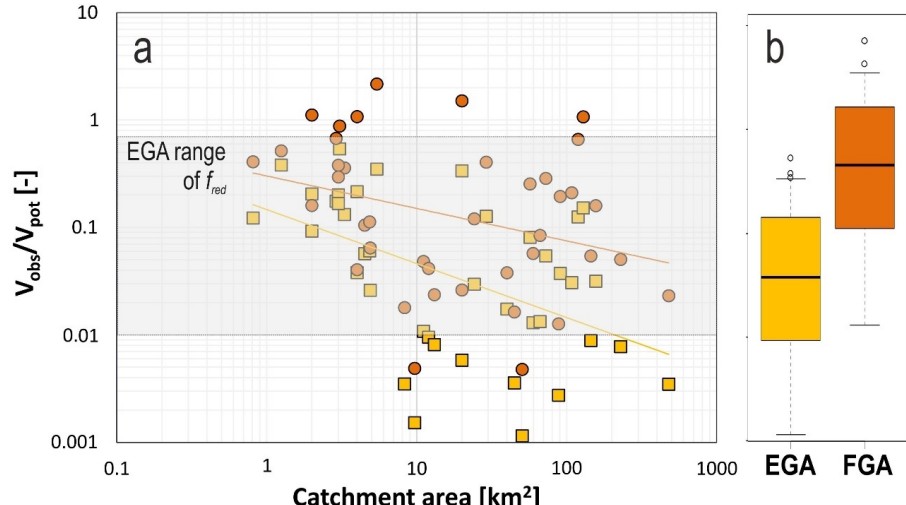


**Figure 10: Ratio of observed wood volumes and potential wood volumes computed by the two**
**models for all sites and their catchment areas. The grey rectangle shows the reduction factor range**
**used for EGA computations.**

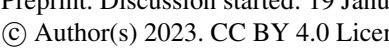



## 5    DISCUSSION

### 5.1    Major differences between the two models and remaining challenges

Both the EGA and FGA are based on a similar general concept, were fed with similar input data (e.g., stream network, forest density, areas affected by landslides and debris flows) and run with defined equivalent scenarios which made the comparison possible. However, there are also some methodological differences that resulted in different model outputs. Here we describe them, while in the following section we discuss our results comparing them to current knowledge and other existing methodologies.

The most relevant difference between the EGA and FGA is the approach to define the areas affected by **bank erosion**, thus the contribution of this recruitment process and the estimated wood supply volumes. EGA uses buffers around the stream network computed for each scenario with one specific width factor, independent of the original channel width. The resulting buffer widths were added on both sides of the original channel width (section 3.3). FGA also assigned scenario-specific buffers, computed with width ratios that vary according to nine channel width classes (Figure S1). Half of the resulting buffer widths were added on both side of the original channel width. As a result, potential bank erosion recruitment areas are generally larger for EGA than for FGA. However, the reduction factors used for the EGA assumed that between 5% and 20% of the potential wood volume within these areas contribute to the estimated wood supply, which resulted in a much lower estimated wood volume. In the case of the FGA, the entire forested area identified as prone to bank erosion along the river network is contributing to wood supply and the volume is reduced based on fuzzy logic pixel values (computed based on sinuosity and channel slope, and going up to 30% of the potential), which resulted in a much larger volume. This difference is particularly relevant for the medium scenario, for which the bank erosion width identified by both models are quite similar, but the resulted wood volumes significantly differ (e.g., average wood volume equal to 114 and 2613 $m^3$ for EGA and FGA respectively for all sites). This difference in the way of computing recruitment areas from bank erosion and related wood volumes explains the second most important difference between the two models. As shown



in section 4.2, landslides are the **dominant recruitment process** in the case of the EGA, whereas
bank erosion is the predominant process in the FGA model. In both models, for landslides and
debris flows, the input data were the trajectories from the SilvaProtect-CH database, but the EGA
applies an expert-based buffer for each scenario to those trajectories, while the FGA groups them
in three classes according to their density. In addition, the fuzzy connectivity applied in the FGA
further reduces the areas identified as prone to mass movements (only for landslides). This
hillslope-channel network connectivity is another methodological difference between the two
models. In EGA, as a proxy for connectivity, only landslide trajectories within 50 m distance from
the stream network were considered. FGA considers connectivity as a function of both the
distance to the channel and the terrain slope (as used by Ruiz-Villanueva et al., 2014c).
Noteworthy, both models use Euclidean distance, but no geomorphometric measures (e.g.,
steepest downslope direction) as often used to assess sediment connectivity (e.g., Cavalli et al.,

2013).

The EGA generally produces much larger potential recruitment areas for landslides and

computes larger wood supplied by landslides than the FGA, for all three scenarios. For the FGA,
landslides are minor supplier of wood for the medium and large scenarios, while their contribution
for the very large volume scenarios significantly increases.

Existing observations show that mass wasting processes, such as debris flows and landslides,

often are the most relevant recruitment processes in smaller headwater catchments (e.g., Rigon et
al., 2012; Hassan et al., 2016; Seo et al., 2010). In contrast, (lateral) bank erosion is often prevalent
farther downstream in larger mountain or lowland rivers, resulting in large volumes of LW supply
by this fluvial recruitment process. This was observed after the large flood in 2005 in Switzerland
(Steeb et al., 2017b), the large flood in the Magra River catchment in Italy in 2011 (Lucía et al.,
2015c; Comiti et al., 2016) and along the Emme river catchment in 2014 (Ruiz-Villanueva et al.,
2018). In smaller streams, bank erosion and channel widening can also be significant, especially
in natural reaches (no stream regulation works), as seen after severe flash floods in Braunsbach,
Germany in 2016 (Lucía et al., 2018). In most of these cases, only a small proportion (<30%) of



the total recruited wood was supplied by mass wasting processes, and the majority of the supply
was due to bank erosion and channel widening along the river network.

Such catchment size-specific trends of dominant recruitment processes are not clearly

prevalent in the model results of EGA and FGA. Generally, the variability in the recruitment
processes and thus in the wood supply is very large, both in empirical data as well as in modelling
results, highlighting the importance of other catchment- and event-specific characteristics. The
relationship of estimated LW supply with catchment characteristics is shown in supplementary
material Figure S2. The highest correlation is seen for forested stream length that can be
interpreted as a proxy for potential supply volume for bank erosion. High correlations also exist
for Melton ratio and relief ratio, both surrogates for watershed slope, a factor that is directly
related to stream power and debris flow and landslide propensity. In general, $V_{obs}$ from EGA
shows slightly higher correlations ($R^2$) with catchments characteristics than FGA. More research
is needed to better understand wood recruitment processes and to improve predictive models on
a physical basis. This will help to determine where and how likely mass wasting (landslides) or
bank erosion could occur.

The results in section 4.3 indicate that there is both under- and overestimation of wood supply

volumes. As shown in Figure S4, potential LW supply $V_{pot}$ generally increases with catchment
size. During a convective storm event, often only a part of the catchment is affected, and therefore
geomorphologically active, so that LW supply may easily be overestimated ($V_{est} > V_{obs}$). In
smaller catchments and torrents, sporadic recruitment processes such as landslides or debris flows
can dominate and deliver large amounts of wood at once, so that wood supply may be
underestimated by our models ($V_{est} < V_{obs}$).

A less relevant difference between the models, and in terms of the total contribution to the

wood volume estimations, is the approach used to assign previously **deposited instream wood**
**loads**. The EGA assignes instream wood load values into three channel width classes (section
3.3), whereas FGA assignes wood load values into three stream order classes (section 3.4). The
main divergence comes from the assumption that the smaller channels contain the largest instream



wood load for the EGA (following observations in 10 small mountain streams in Switzerland
from Rickli et al., 2018), while the FGA assumes that larger loads are present in medium order
channels (as proposed by Wohl, 2017). Despite the different approaches, both models used
empirical data from Ruiz-Villanueva at al. (2016) to assign volumes, and the resulting wood load
volumes were only significantly different in the case of the medium scenario (Figure 7).

These differences in the methodologies result in differences in the outcomes, in terms of the

**potential and estimated wood supply**. The EGA generally produced larger potential recruitment
areas. The volume reduction factors applied in EGA are, however, on average much smaller than
the respective fuzzy-logic values created in FGA (Figure S3). As a result, estimated wood supply
is generally larger for FGA, as shown in section 4.1. For our test catchments, the application of
simple empirical volume reduction factors as part of the EGA model has proven to be similarly
accurate in estimating LW volumes, in comparison with a spatially explicit approach such as the
FGA model. Still, both the expert-based buffer widths and the reduction factors were defined for
the test catchments and validated for similar catchments located in the Alps and pre-Alps, and so
they should be carefully tested if applied to other rivers with different characteristics. The fuzzy
logic approach indirectly includes this uncertainty or imprecise information (i.e., buffer widths
and volume reduction factors), and allows being computed without prior existing observations or
knowledge. In both cases, the two models may over- or underestimate the wood volumes, but
allow reliable computation of wood supply volumes at the catchment scale and for three scenarios.

**5.2    Qualitative comparison of EGA and FGA with other similar**

**approaches**

As described in the introduction, just a few approaches have been proposed to compute

wood supply at the catchment scale considering different recruitment processes (e.g., landslides,
debris flows, bank erosion). As those models presented here, most frameworks, particularly those
based on GIS and geoprocessing (e.g., Mazzorana et al., 2009) do not attempt to simulate the
actual recruitment processes, but they used existing information on areas susceptible to certain



processes (as the EGA and FGA) from hazard maps or other sources or apply expert-based buffers
(as the EGA). Very few models simulate only one recruitment process (i.e., landslides or bank
erosion) explicitly (Lucía et al., 2015a; Cislaghi et al., 2018; Zischg et al., 2018; Franceschi et al.,
2019; Gasser et al.; 2018, 2020). Yet, a model that simulates coupled processes to compute wood
supply is still lacking. In existing approaches, physically based models are combined with
empirical approaches to identify recruitment areas from one single process and compute wood
supply at the catchment scale. Still, these models require additional input data, such as
precipitation, discharge, soil characteristics etc., which is usually not available or challenging to
obtained at the desired resolution. In addition, they are much more expensive in terms of
computational time, which limits their application to larger areas. Therefore, there is a gap
between the current state-of-the-art of geomorphic process modelling and wood recruitment and
supply estimation.

Moreover, the majority of existing models used to predict wood supply are deterministic,

in that they do not consider the natural process variability and parameter uncertainties. Only the
fuzzy logic approach (Ruiz-Villanueva et al., 2014c; Ruiz-Villanueva and Stoffel, 2018)
indirectly considers uncertainty, but it does not represent a description of the physical supply
processes. A few stochastic models have been proposed (e.g., Bragg, 2000; Eaton et al., 2012;
Gregory et al., 2003) to simulate wood recruitment, but they were designed to work at the scale
of the river reach only. At the catchment scale, a probabilistic multi-dimensional approach has
recently been proposed (Cislaghi et al., 2018) to study wood sources from hillslopes, modelling
areas susceptible to landslides, but it neglects other processes such as bank erosion. The latter
process has been considered in one of the most recent studies on LW (Gasser et al., 2020).

On the other hand, empirical estimation formulas (e.g., Steeb, 2018; Rickenmann, 1997;

Uchiogi et al., 1996) are easier and faster to apply to estimate LW supply. However, they provide
only an estimate for the whole catchment under investigation, without any spatial differentiation.
EGA and FGA, on the other hand, support a comprehensive spatial overview and direct attention
to areas in which a more precise assessment of the instream wood situation is necessary, e.g.,





through field surveys or expert assessments. Figure S5 shows that the EGA and FGA modelling
results approximately correspond to the 50-90% relation between $V_{obs}$ and catchment area as
described with the empirical formula of Steeb (2018).

### 5.3    Uncertainty in the observed and modelled LW volumes

The two GIS approaches presented here yielded similar orders of magnitude of LW supply

for a given catchment and for the three designed scenarios. Still, several uncertainties associated
with the estimation of LW supply remain, and they are not just related to the obtained results and
the applied methodologies, but also to the available observations (coming from surveys after flood
events) used for calibration and validation.

The observed wood volumes $V_{obs}$ were compiled mostly from technical reports of post-

event analyses, and these values might be in some cases only rough estimates, with a considerable
uncertainty. LW volumes were estimated based on LW deposits and piles in the field, for which
the volume and the corresponding wood content (or pore volume, respectively) must be estimated.
The assessment of the wood volume of such accumulations might be challenging and uncertainty
might be high (Spreitzer et al., 2020; Thevenet et al., 1998). Some of the observed wood volumes
$V_{obs}$ were also determined on the basis of forest loss areas, for which a pre-event forest density
value $W$ must be assumed. In the analysis made with the GIS models, the forest density raster
map of Ginzler et al. (2019) was used, which may differ from values used during the post event
surveys. Furthermore, the time gap between a LW transporting flood event and the survey year
on which the forest density map is derived from, needs to be accounted for. Depending on this
relationship, wood volumes may be underestimated (i.e., survey year after flood event) or
overestimated (i.e., survey year before flood event). This circumstance could also explain why in
some cases of the FGA calculations the potential wood volume is even smaller than the observed
wood volume ($V_{obs}/V_{pot} > 1$; see Figure 10a). This discrepancy appeared mostly in one large
catchment (i.e., Chirel) and its subcatchments (i.e., Fildrich, Goldbach, Rütigrabe), and could be



related to the forest density data used to compute the wood supply volumes, which was computed
with the forest after the large flood in 2005.

The observations we used remain a unique and extensive dataset (Steeb et al., 2019a),

which allowed us to parametrize the models more accurately. The EGA uses empirical volume
reduction factors that were derived from this dataset for the conversion of $V_{pot}$ to $V_{est}$. In case of
debris flows, for example, the volume reduction factors $f$ also rely on an event analysis of the
August 2005 flood in Switzerland by Rickenmann et al. (2008), who showed that, on average, 11-
19 % of all torrents in the main investigated mountain river catchments were associated with
debris flow activity. This percentage range was used to define the reduction factors as shown in
Table 2. This highlights the importance of in-depth post flood event analyses, as these provide
valuable empirical datasets that can be used to validate and further develop models to estimate
supplied LW volumes. The application of models should not replace field work surveys, but they
should be used in a complementary manner.

Another source of uncertainty is given by the SilvaProtect-CH trajectories. Since their input

data, in particular geology, provide a large-scale representation of natural conditions (see text in
the supplementary material), the SilvaProtect-CH trajectories are best suitable for use on a
catchment-scale range. Furthermore, SilvaProtect-CH trajectories generally result in a pessimistic
picture under unfavourable conditions (e.g., no consideration of the stabilizing influence of
vegetation cover). As a consequence, only a small part of the trajectories is expected to be active
during rainfall and consequent floods. In addition, the actual run-out zones of mass wasting
processes may often be shorter than the modelled trajectories.

One important limitation of the EGA and FGA models presented in this study is that the

available input forest cover, does not provide any further information about the forest typology,
structure and species composition. Despite the role that differences in forest may play in
stabilizing the soil and slopes and in influencing bank erosion and hillslope stability (Gasser et
al., 2019), the two methods do not explicitly consider this effect. Moreover, the type, structure
and stage of forest stand control the extent to which trees can be uprooted and recruited and





supplied to rivers (Mazzorana et al., 2009; Ruiz-Villanueva et al., 2014c). This aspect was
described as the vegetation resistance defined by Ruiz-Villanueva et al. (2014c) based on the tree
species and forest stage, the structural classification of forested areas made by Blaschke et al.
(2004) and the availability indicator used by Mazzorana et al. (2009). However, there was no
available information with the required spatial resolution to consider the proportion of different
species, the stage (e.g., remnant or reforested) or the age of the forest stand. Neglecting the
different response of different forest types may result in an overestimation of supplied volumes.

As discussed above, modelling and quantification of wood supply volumes is characterised

by many uncertainties. After all, the two models presented in this study allow quantifying the
magnitude of the expected LW supply, thus further expert judgement and knowledge of local
(geomorphic) characteristics is required to adequately interpret the results. The ratio between
predicted and observed LW volumes varies by about 1-2 order of magnitudes. For comparison it
is noted that a similar or even larger range of uncertainty can be expected for the estimation of
bedload volumes transported during floods (e.g., Rickenmann and Koschni, 2010).

**5.4    Implications for hazard assessment and river management**

From a practical perspective, geospatial LW modelling results can be used for hazard

assessment, infrastructure design, and the definition of management strategies. From a scientific
perspective, further applications are possible. For example, estimated wood volumes can be
applied as an input for a wood transport model, such as Iber-Wood (Ruiz-Villanueva et al., 2014a,
2014b, 2015) or other approaches (e.g., Mazzorana et al., 2011), to define realistic boundary
conditions. Furthermore, if no observation data are available for reference, estimated wood
volumes from EGA and FGA can be used to quantify blocking probabilities due to LW at bridge
piers or at other critical cross-section (Schalko, 2019; Schalko et al., 2018; Schmocker and
Weitbrecht, 2013).





As described in section 4.2, the average proportion of instream deadwood (instream wood
load) from the total potential LW supply in the 40 test catchments ranged between 2-13 % (Figure
6). This range is confirmed by other studies and event analyses (Dixon, 2013; Rickli et al., 2018;
Waldner et al., 2009). It can be concluded that instream deadwood generally accounted for only
a small proportion of the total LW transported during past floods in Switzerland. Rather, it is
freshly recruited wood that made up the majority of the transported wood volumes. Deadwood
alone, both on the forest floor and in the channel itself, may therefore only lead to a limited
increase in risk from a natural hazard management perspective. As a consequence, the artificial
removal of deadwood from the stream and its surroundings is not always necessary, keeping in
mind the ecological benefits of instream wood.
EGA and FGA are area-wide products that can be applied in any Swiss catchment. They
use a standardized procedure and nationwide homogeneous data, which facilitates a comparison
between catchments. The methodology is flexible and can be adapted to other regions outside
Switzerland if recruitment processes (especially with regard to SilvaProtect-CH trajectories) were
modelled with more generic approaches.
The two models presented here correspond to a hazard index mapping in terms of
processing depth and degree of detail for a hazard assessment. The geospatial modelling results
indicate areas of potential LW recruitment, however without precise information about the
intensities occurring. In contrast, the estimated LW supply for the large scenario is based on the
data of events with a return period of approximately 50 to 150 years. The approach presented here
is a useful tool to give a comprehensive overview and direct attention to areas where a more
precise assessment of the LW situation is probably useful, for example in connection with an
estimation of sediment loads in torrents.



## 6   CONCLUSIONS

Two GIS-based models are presented in this contribution to identify large wood (LW) sources and to estimate LW supply to rivers. Both models, called empirical GIS approach (EGA) and Fuzzy-Logic GIS approach (FGA), consider landslides, debris flows, bank erosion, and mobilization of instream wood as recruitment processes. The results are volumetric estimates of LW supply based on three different scenarios of process frequency and magnitude. Results of model applications to 40 Swiss catchments were used to compare both the two models with each other and the performance in relation to observed (empirical) LW volumes. Further, a literature review of existing LW supply models proposed in the last 35 years was conducted, set into context and remaining challenges were identified.

EGA shows significantly higher values for potential LW supply. However, after reducing the potential volume with different methods, estimated LW supply volumes are in the same order of magnitude for both models, with FGA showing generally somewhat larger values. In case of EGA, landslides are the dominant recruitment process, whereas bank erosion is dominant for FGA. Both models show under- and overestimation of observed wood volumes $V_{obs}$, with more tendency for overestimation. Overestimation stays generally within two orders of magnitude (typically larger values for FGA), underestimation within on order of magnitude (typically smaller values for EGA).

The modelling and quantification of wood supply volumes is characterised by many uncertainties. After all, the two models presented in this study allow quantifying the magnitude of the expected LW supply, thus further expert judgement and knowledge of local (geomorphic) characteristics is required to adequately interpret such results. LW supply modelling can be further improved by integrating more physically-based and/or probabilistic inputs for the spatial identification of recruitment processes. Likewise, the parametrization and validation of LW supply models remain complex. Post flood event analysis provide valuable empirical datasets that can be used to validate results and further develop LW supply models that can be useful for hazard assessment, infrastructure design, and the definition of management strategies.





**ACKNOWLEDGEMENTS**
We thank the Swiss Federal Office for the Environment (FOEN) for funding the research
program "Large Wood Management in Rivers" (WoodFlow research program; contract no.
15.0018.PJ/O192-3154).
Special thanks go to Peter Waldner (WSL) for providing valuable empirical data from flood
events; Benjamin Kuratli (formerly University of Zurich) for helping to develop earlier versions
of the EGA; Bronwyn Price, Christian Ginzler and Markus Huber (all WSL) for providing data
from the Swiss National Forest Inventory; and finally, Stéphane Losey (FOEN) for providing all
the required SilvaProtect-CH data.



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
