# Peer review of "Geospatial modelling of large wood supply to rivers: a state-of-the- art model comparison in Swiss mountain river catchments"

_Earth Surface Dynamics, 2022_

## Referee Comment (RC1)

**Review of "Geospatial modelling of large wood supply to rivers: a state-of-the-art model comparison in Swiss mountain river catchments"**

**1. Summary**

This paper presents an overview of existing modelling approaches for estimating LW supply to rivers, and a more detailed comparison of two GIS-based approaches including catchment characteristics and estimated LW supply (Steeb et al., 2019a) for 40 Swiss catchments for 50 to 150-yr return period events. The empirical potential LW volume reduction factors used in one model (EGA) were obtained from the observational dataset, reducing the contribution as a model validation. In general, as the authors acknowledge, the uncertainty in both observed and modelled LW supply is large (Figure ), and both modelling approaches perform similarly for LW supply estimation, with apparent overlap between the range of model results and the range of field data. The authors discuss aspects contributing to uncertainty in the modelled and observational results.

In general the paper seems caught between two directions, on the one hand a review paper examining the range of modelling approaches, and on the other a comparison of two GIS-based approaches through use of a detailed observational dataset. The authors state that "the approach presented here is a useful tool to give a comprehensive overview and direct attention to areas where a more precise assessment of the LW situation is probably useful." I appreciate that the authors acknowledge the large degree of variation in both observational and modelled results, which is interesting and presents opportunities to strengthen the contribution of this manuscript. Do the main sources of observational uncertainty correspond to the main sources of uncertainty in both models? For each model, which steps have the highest contribution to final uncertainty? Is this broadly the same across all catchments, and could this be helped by future field observations? How does the present degree of uncertainty impact end use in hazard assessment—could this be improved if the models and observations were refined, or is the present degree of variation acceptable?

**Line-by-Line comments:**

Line 53: Please add references for this statement ("The main damage potential…").

Line 64: To improve the flow of this sentence, please incorporate the parenthetical list in to a main stentence structure.

Lines 69-75: Here, additional context would be helpful to connect the model comparison and model review portions. Why the focus on these two GIS-based approaches?

Lines 112-141: To contextualise model development it would be helpful to incorporate a short overview of accompanying technological advances in GIS and LiDAR enabling current model development.

Line 163 (approx.): There is a broad range of models under comparison, making Table 1 difficult to follow. A schematic diagram illustrating different modelling approaches would be helpful, perhaps with key advantages and/or disadvantages of the scheme classes.

Table 1: it would be helpful if line numbers were included and used in the text to reference specific models to easily find entries in the table. If possible could a column specifying calibration datasets and/or model validation procedures would be helpful in understanding progression of model-data comparisons and highlight areas for future work later in the text.

Line 171: Here as above in the Introduction, additional context on why the two GIS-based approaches were chosen for comparison and relationship to the reviewed model development would be helpful in contextualising the comparison portion of the manuscript.

Figure 1: The quantitative definition of 'medium, large, very large' events is not presented until later in the text; please could this be defined earlier (or mentioned in the figure caption) so that the figure can be easily interpreted.

Lines 197-198: Here, more information on the calibration dataset would assist in interpreting the results. This information Is currently largely presented in Section 5.3. However, the degree of variation in both the observational and modelled results is interesting and an examination of the reasons for variation and accompanying areas for future work in both approaches could be very helpful in motivating future work to fill identified research gaps.

Figure 2: This figure does not seem to add significant support to the text, especially in light of repeated reference to Supplemental Information figures. Could this figure be replaced by either Figure S1A

In this section more fully present the difference between the EGA buffer and FGA fuzzy-logic approaches would be helpful here. The difference in approach is a key focus of the Results, but equations are not included and the procedure details are not yet fully brought out in the text or figures.

Lines 283-286: More details of the choice of $f$ are needed here. What range did the literature review yield? Can a more in-depth discussion of the five catchment mentioned be examined? Or, if these details are already in previous literature, please clarify.

Figure 5: From Lines 355-356 I believe that the observed data corresponded to the 'large' event scenario. This should be made clear in the figure and figure caption, and in Table 3.

Lines 404-405, and elsewhere: In general the Results would benefit from stronger quantitative comparison to contextualise results for the reader. Here, it would be helpful to discuss the variation in results in addition to mean values, and elsewhere (ex. Lines 421-424, 483, 486, and elsewhere) quantitative assessment of what constitutes "larger" and "much larger" is needed to interpret the Results.

Figure 9: Please label the lines of equality (y=x) in the subfigures.

Figure 10: Please explain the plotted gray lines.

Discussion: The Discussion and overall contribution of this manuscript would be strengthened through a more in-depth discussion of the observed variation in model and observational results presented in the Results (please see second paragraph of Summary). It is mentioned (ex. Lines 739-740) that the observational dataset enables comparison between catchments. Perhaps a catchment comparison case study be highlighted to help demonstrate differences in the models or variation in observational data?

---

## Author Comment (AC1)

**CC1**: 'Comment on esurf-2022-69', Andrés Iroumé, 21 Feb 2023 reply

I am really enjoying this manuscript.

Thank you for taking the time to read our manuscript and make comments. Here we provide detailed responses and indications of where we edited the text.

Some general comments:

1. In page 1, lines 31-33, the authors write: "Regardless of the limitations of these models, they proved extremely useful for hazard assessment, and the design of infrastructure and other management strategies". Is this a result from this ms, or the models had been used for hazard assessment?

Yes, the models had been used already by practitioners for some engineering applications. Still, we must stress some limitations regarding the use of the models. One important limitation encountered by some practitioners is the use of licensed software, as both models were developed in ESRI software and need some advanced licenses that might not be always accessible to private companies. Future developments may consider the migration to open-source software. Another limitation is regarding some input data (i.e., SilvaProtect trajectories) that was provided by the Swiss Federal Office for the Environment and is not publicly available to private companies. However, this data could be still requested by the practitioners. This, however, is still a limitation to applying the models outside Switzerland, as this data set is only available for the Swiss territory.

We expanded the discussion in section 5.4 to account for these limitations.

In addition, how accurate can they be to assess hazard in particularly because, as the authors write, they provide different outcomes both in terms of LW volumes and LW sources.

For hazard assessment in river engineering, the identification of a specific dominant recruitment process may not be the most relevant aspect, but having an estimate of LW volumes is helpful for example to properly dimension infrastructures. For forest management (LW hazard prevention), however, the identification of the recruitment process is important. Therefore, a higher resolution may be required (e.g., Gasser et al. 2020, with root reinforcement), or overlapping recruitment areas as an indicator. Chapter 5.4 further discusses practical implications.

2. Page 4, lines 67-68. About the data scarcity. Perhaps there are more recent publications?

Yes, we added the most recent references:

Gurnell AM, Bertoldi W. 2020. Wood in Fluvial Systems. 2nd ed. Elsevier Inc. Editor(s): John (Jack) F. Shroder, Treatise on Geomorphology (Second Edition), Academic Press, 2022,

Pages 320-352, ISBN 9780128182352http://dx.doi.org/10.1016/B978-0-12-409548-9.12415-7

Nakamura F, Seo J Il, Akasaka T, Swanson FJ. 2017. Large wood, sediment, and flow regimes: Their interactions and temporal changes caused by human impacts in Japan. Geomorphology 279: 176–187. DOI: 10.1016/j.geomorph.2016.09.001. http://linkinghub.elsevier.com/retrieve/pii/S0169555X16308078

Wohl E et al. 2019. The natural wood regime in rivers. BioScience 69: 259–273. DOI: https://doi.org/10.1093/biosci/biz013

3. Page 4, lines 83-84. The existing approaches are the published approaches? Seems a redundance.

We specified the sentence as follows:

"Here we compile information on approaches and expand these previous overviews to provide an updated review of published approaches to model recruitment processes and to quantify LW supply"

4. Page 5, line 115, about source area. The areas can source LW to the streams if they are connected to the streams. The issue of connectivity is little addressed, here and along the text. Page 6 line 131 and page 7 line 151, the issue of connectivity is briefly presented. As an example, in page 13, lines 195-196 the authors refer to volume reduction factors. I assume that connectivity is a driving issue. Please comment.

How connectivity is handled within the two models, is described on lines 546-554 (original ms).
FGA considers connectivity as a function of both the distance to the channel and the terrain slope. EGA is simpler, using buffer widths for both process trajectories and stream network. Connectivity is also implicitly considered using the Silvaprotect-CH trajectories as they represent flow paths and runout distances of debris flows and landslides (see supplementary material).

5. Page 15, line 239. The revised literature here, is from Swiss channels?

The revised literature is mostly, but not exclusively from Switzerland.

Regarding the EGA, details are provided in Lines 264-270 (original ms): data from ten rivers in Swiss torrents.

Regarding the FGA, we added to the text in Lines 344-347 (original ms):

"This was estimated by assigning wood load values reported for European mountain rivers in the literature (Ruiz-Villanueva et al., 2016) to the different river segments grouped by stream order classes (following Wohl, 2017)…."

Wohl E. 2017. Bridging the gaps: An overview of wood across time and space in diverse rivers. Geomorphology 279: 3–26. DOI: 10.1016/j.geomorph.2016.04.014.

6. In general, the two models identify different dominant processes as wood sources (landslides by the EGA, and bank erosion by the FGA model). So, can they really be comparable? Perhaps one is more suited to smaller headwaters, and the other to larger rivers? Please comment.

We believe that the two approaches are comparable. They used similar input data, but source areas and reduction to estimated wood supply were calculated differently. There are differences in dominant recruitment processes, but the estimated wood supply volumes were still similar in magnitude and trends. For example, under- and overestimation depending on catchment size behave the same way for EGA and FGA (see Fig. 9). Given that the two models use the same input data but identify different dominant processes reflects the fact that it is difficult to (spatially) predict the geomorphic processes that contribute to wood supply in the studied catchments. This aspect is deeply discussed in the discussion section.

**Citation**: https://doi.org/10.5194/esurf-2022-69-CC1

---

## Author Comment (AC2)

**Review of "Geospatial modelling of large wood supply to rivers: a state-of-the-art model comparison in**

**Swiss mountain river catchments"**

**1. Summary**

This paper presents an overview of existing modelling approaches for estimating LW supply to rivers, and a more detailed comparison of two GIS-based approaches including catchment characteristics and estimated LW supply (Steeb et al., 2019a) for 40 Swiss catchments for 50 to 150-yr return period events. The empirical potential LW volume reduction factors used in one model (EGA) were obtained from the observational dataset, reducing the contribution as model validation. In general, as the authors acknowledge, the uncertainty in both observed and modelled LW supply is large (Figure ), and both modelling approaches perform similarly for LW supply estimation, with apparent overlap between the range of model results and the range of field data. The authors discuss aspects contributing to uncertainty in the modelled and observational results.

In general, the paper seems caught between two directions, on the one hand, a review paper examining the range of modelling approaches, and on the other a comparison of two GIS-based approaches through the use of a detailed observational dataset. The authors state that "the approach presented here is a useful tool to give a comprehensive overview and direct attention to areas where a more precise assessment of the LW situation is probably useful." I appreciate that the authors acknowledge the large degree of variation in both observational and modelled results, which is interesting and presents opportunities to strengthen the contribution of this manuscript.

Do the main sources of observational uncertainty correspond to the main sources of uncertainty in both models? For each model, which steps have the highest contribution to final uncertainty? Is this broadly the same across all catchments, and could this be helped by future field observations? How does the present degree of uncertainty impact end use in hazard assessment—could this be improved if the models and observations were refined, or is the present degree of variation acceptable?

We thank the reviewer for the positive comments, suggestions, and detailed review. We addressed each of these important questions here and in the revised manuscript.

**Line-by-Line comments:**

Line 53: Please add references for this statement ("The main damage potential…").

We edited the sentence as follows: "The associated damage potential of LW may depend, among other variables, on the volume of transported LW (Mazzorana et al., 2018)."

Mazzorana B, Ruiz-Villanueva V, Marchi L, Cavalli M, Gems B, Gschnitzer T, Mao L, Iroumé A, Valdebenito G. 2018. Assessing and mitigating large wood-related hazards in mountain streams: recent approaches. Journal of Flood Risk Management 11: 207–222. DOI: 10.1111/jfr3.12316

Line 64: To improve the flow of this sentence, please incorporate the parenthetical list into a main sentence structure.

The sentence has been edited accordingly:

"The estimation of exported wood involves many uncertainties that are difficult to quantify, because LW transport happens at the end of a long process cascade, usually starting with precipitation as a trigger, followed by a flood formation and the occurrence of recruitment processes as wood suppliers, and the increased discharge as a transport medium."

Lines 69-75: Here, additional context would be helpful to connect the model comparison and model review portions. Why focus on these two GIS-based approaches?

We provided additional information in the revised manuscript (at the end of the introduction):

"This work reviews the state-of-the-art in wood supply modelling and presents a comparison of two recent GIS-based approaches developed in the context of a research-applied project funded by the Swiss Federal Office of the Environment. The literature review provides an updated compilation of published approaches to model recruitment processes to quantify LW supply, clarifying the approaches by model type and summarizing their main characteristics, such as processes considered, and their temporal and spatial scales. We then focus on two GIS-based models that were developed based on a similar general concept, used similar input data and were applied to the same study sites. Despite their similarities, the models differ in some respects and result in somewhat different outcomes. These differences are used to stress the limitations and strengths of the two models, to compare them with other recent approaches included in the literature review and to discuss uncertainties and challenges related to the modelling of LW supply. In addition, we also consider implications for flood hazard assessment and river management."

Lines 112-141: To contextualise model development it would be helpful to incorporate a short overview of accompanying technological advances in GIS and LiDAR enabling current model development.

We edited lines 112-115 (original ms) as follows:

"The rapid proliferation of remote sensing and the advances in computing sciences and geographic information systems (GIS; Bishop and Giardino, 2022) resulted in the development of another group of models (i.e., geospatial models). These GIS-based models allow a spatially explicit assessment of different LW recruitment processes, the identification of source areas and the estimation of LW volumes, expanding the analysis to larger areas, covering multiple (sub-)catchments."

Bishop, M.P., Giardino, J.R. 2022. Chapter 1.01 - Technology-Driven Geomorphology: Introduction and Overview. In: Editor(s): John (Jack) F. Shroder, Treatise on Geomorphology (Second Edition), Academic Press, 2022, Pages 1-17, ISBN 9780128182352, https://doi.org/10.1016/B978-0-12-818234-5.00171-1. (https://www.sciencedirect.com/science/article/pii/B9780128182345001711)

Line 163 (approx.): There is a broad range of models under comparison, making Table 1 difficult to follow. A schematic diagram illustrating different modelling approaches would be helpful, perhaps with key advantages and/or disadvantages of the scheme classes.

Thanks for the suggestion. We edited the table, and hopefully made it clearer and easier to follow.

Table 1: it would be helpful if line numbers were included and used in the text to reference specific models to easily find entries in the table. If possible could a column specifying calibration datasets and/or model validation procedures would be helpful in understanding progression of model-data comparisons and highlight areas for future work later in the text.

We included the numbering and used it in the text when needed. However, we did not add information about calibration datasets or validation procedures as that was not available in many cases. We provide the source references for further information on each case and suggest reading the original papers.

Line 171: Here as above in the Introduction, additional context on why the two GIS-based approaches were chosen for comparison and relationship to the reviewed model development would be helpful in contextualising the comparison portion of the manuscript.

We explain this in Lines 176-178 (original ms), and now also added the context in the last paragraph of the introduction.

Figure 1: The quantitative definition of 'medium, large, very large' events is not presented until later in the text; please could this be defined earlier (or mentioned in the figure caption) so that the figure can be easily interpreted.

We added a definition of the scenarios in the caption of Figure 1 and referred to the text that describes them in detail.

Lines 197-198: Here, more information on the calibration dataset would assist in interpreting the results. This information Is currently largely presented in Section 5.3. However, the degree of variation in both the observational and modelled results is interesting and an examination of the reasons for variation and accompanying areas for future work in both approaches could be very helpful in motivating future work to fill identified research gaps.

The uncertainties and challenges related to the model calibration are discussed in detail in section 5.3. However, we added a sentence at the end of section 3.1 to give more information about the calibration dataset:

"Modelling results were validated by comparison with available empirical data documented after flood events in Switzerland (Steeb et al., 2021, 2022). This dataset documents recruited and transported quantities of large wood together with the associated catchment and flood-specific parameters, including associated recruitment processes (Table S1 in supplementary material)."

Steeb N, Rickenmann D, Rickli C, Badoux A. 2021. Large wood event database. EnviDat. https://www.envidat.ch/dataset/large-wood-event-database

Figure 2: This figure does not seem to add significant support to the text, especially in light of repeated reference to Supplemental Information figures. Could this figure be replaced by either Figure S1A In this section more fully present the difference between the EGA buffer and FGA fuzzy-logic approaches would be helpful here. The difference in approach is a key focus of the Results, but equations are not included and the procedure details are not yet fully brought out in the text or figures.

The wood stock raster is an important input to the models, and since it is not visualized elsewhere in the manuscript, we prefer to keep it as it is in section 3.2.3. Figure S1A is a FGA-specific graph with a level of detail that is better suited for the supplementary material, in our opinion. The methodological differences between EGA and FGA are described in detail in the following sub-sections 3.3 and 3.4, including the classification of buffer widths (no equations required).

Lines 283-286: More details of the choice of *f* are needed here. What range did the literature review yield? Can a more in-depth discussion of the five catchment mentioned be examined? Or, if these details are already in previous literature, please clarify.

We further specified in the text that the five calibration catchments are from the well documented 2005 flood event. In the caption of Figure 3 it is stated that the five catchments were used to calibrate the volume reduction factors from the EGA approach so that the estimated supplied wood was in the same order of magnitude as the observed values from past flood events. A reference is added to the caption for further details (Steeb et al., 2019b).

We forgo, however, to explicitly specify ranges of *f* values from literature in the manuscript. We doubt that this would be really helpful, since the potential recruitment areas may be defined rather differently in different studies. Direct comparison with our *f* values is therefore not given.

Figure 5: From Lines 355-356 I believe that the observed data corresponded to the 'large' event scenario. This should be made clear in the figure and figure caption, and in Table 3.

Yes, in general, the observed data correspond to the 'large' event scenario. We added a sentence accordingly in the caption of Figure 5 and Table 3:

"Observed refers to the reported LW volumes after flood events, in most cases equivalent to the large scenario."

Lines 404-405, and elsewhere: In general the Results would benefit from stronger quantitative comparison to contextualise results for the reader. Here, it would be helpful to discuss the variation in results in addition to mean values, and elsewhere (ex. Lines 421-424, 483, 486, and elsewhere) quantitative assessment of what constitutes "larger" and "much larger" is needed to interpret the Results.

Lines 421-424 (original ms): We added a paragraph to section 4.1 describing the variation between the models, including quantitative values. We also added values of standard deviation (SD) and root mean square error (RMSE) of the model outputs to Table 3. References are made to Table 3 and Table 4, where quantitative values are shown. The newly created Figure S6 in the supplementary material further visualizes and quantifies the comparison of EGA and FGA outputs.

Line 483 and 486 (original ms): Quantitative assessment is stated with the order of magnitude of variation.

Figure 9: Please label the lines of equality (y=x) in the subfigures.

It is explained in the revised caption of the figure.

Figure 10: Please explain the plotted grey lines.

This is explained in the revised caption: The grey rectangle shows the reduction factor range used for EGA computations.

Discussion: The Discussion and overall contribution of this manuscript would be strengthened through a more in-depth discussion of the observed variation in model and observational results presented in the Results (please see second paragraph of Summary). It is mentioned (ex. Lines 739-740) that the observational dataset enables comparison between catchments. Perhaps a catchment comparison case study be highlighted to help demonstrate differences in the models or variation in observational data.

We added a paragraph in section 4.3 discussing the differences in the distribution of observed vs. estimated wood supply, including reference to a new Table S3 and a new Figure S7 in the supplementary material.

A comparison case study was made during the WoodFlow research program (FOEN, 2019), and we added a reference at the end of line 740. But it would go beyond the scope of this manuscript to present the case study here.

---

## Author Comment (AC3)

**Review of "Geospatial modelling of large wood supply to rivers: a state-of-the-art model comparison in Swiss mountain river catchments"**

**Reviewer 2**

Dear Authors, I have really enjoyed reading your ms, which I think represents a very valuable piece of work for improving the quantitative estimation - and most importantly the knowledge about related uncertainties - of wood fluxes during flood events. In my opinion, the text is generally clear enough and figures/Tables quite informative. I just have a few suggestions to improve the work, as listed below:

*Response: We thank the reviewer very much for the time to revise the manuscript and the positive comments that helped us to improve our work. We replied to each comment and made the changes in the text accordingly.*

- in the description of the FGA, I think it would be useful for the reader to have the relationship between the max distance and slope steepness for the landslide "connectivity" assessment. At the moment it says just "further away if slope >40% (line 318). But how far upslope? this is an important parameter, I believe.

*Response: we clarified this point as follows:*

*Lines 315-318 (original ms) "Trees located in a landslide-prone pixel or within the toppling influence area (defined as a buffer equal to two times the tree high; here 100 m) may reach the channel if they were close enough (Euclidean distance to channel network < 50 m), or further away (Euclidean distance up to 100 m) but on a steep slope (>40%)."*

*This is discussed in lines 546-554 (lines in original ms).*

- for both EGA and FGA please make explicit whether for bank erosion the erodibility of the lateral channel boundaries are considered or not. In other terms, can the models exclude (or greatly limit) bank erosion inputs in case of bedrock banks? If I am not wrong this is not the case, and thus I suspect bank erosion contribution may be overestimated, especially in steep reaches by FGA as the use of the width ratio is related to channel steepness, but steep channels may be extensively bordered by stable bedrock areas. I think the authors may wish to discuss how the introduction of erodibile vs non-erodible areas for both models may improve the performance of models. Such an approach was introduced by Franceschi et al. (2019) in combination with the use of width ratios (whose values can be set by the users)

*Response: The erodibility of the channel boundaries was not considered in the models. The reviewer is right, that this assumption results in an overestimation of the wood supplied by bank erosion. We discuss this aspect in the discussion (added after line 540):*

*"Moreover, the erodibility of the channel boundaries was not considered in the models. Anthropogenic elements like bank protections, check-dams, and bridges or the presence of bedrock may limit bank erosion and widening, and thus wood supply. This information was not available at the required resolution and spatial scale for the analysed river catchments, and thus we could not include it. This also results in an overestimation of the computed wood volumes by bank erosion, which could be more relevant in the FGA than in the EGA (for which the volume reduction coefficient could be more easily adjusted)."*

- Unfortunately, Franceschi et al (2019) is not published yet, as you know. However, please not that the preprint has a longer list of authors that those reported in the references. Please check it here "https://www.researchgate.net/publication/330397920_GIS-based_approach_to_assess_large_wood_transport_in_mountain_rivers_during_floods".

*Response: we corrected the reference:*

*Franceschi S, Antonello A, Lucia Vela A, Cavalli M, Crema S, Tonon G, Comiti F. 2019. GIS-based approach to assess large wood transport in mountain rivers during floods. Preprint DOI: 10.13140/RG.2.2.31787.08480*

Also note that this model accounts for both landslides and bank erosion inputs, differently from what reported at lines 625.

*Response: We removed Franceschi et al (2019) from line 625, and edited the paragraph as follows:*

*Lines 624-627: "Most existing models simulate only one recruitment process (i.e., landslides or bank erosion) explicitly (Lucía et al., 2015a; Cislaghi et al., 2018; Zischg et al., 2018; Gasser et al.; 2018, 2020), and a few consider mass movements and fluvial processes (e.g., Franceschi et al.,2019). Yet, a model that simulates coupled processes to compute wood supply is still lacking."*

In addition, Franceschi et al model is capable to work with single trees and thus provide statistics about number of elements and their size (For table 1).

*Response: we added the following to line 706 (original ms):*

*"Unlike in the approach used by Franceschi et al (2019) or Gasser et al., (2018), who detected individual trees from high-resolution LiDAR data, in our case, there was no available information with the required spatial resolution to consider the dimensions, proportion of different species, or the stage (e.g., remnant or reforested) or the age of the forest stand."*

- The authors should comment about the need/benefit for hazard prediction to model - in addition to the wood supply presented here - the propagation of LW in channels during floods (1D or 2D, including critical sections as bridges where most LW can get trapped), I think that especially for the larger basins the lack of propagation (and thus the longitudinal disconnectivity) may be partially responsible for the important overestimations showed in this study. I hope you will find my comments of use. Best wishes! Francesco Comiti

*Response: The aspect related to the overestimation of wood supply and the size of the catchment is clearly shown in Figures 9 and 10, and discussed in lines 585-590. We added some lines to refer to the lack of transfer or propagation in our models:*

*"Another important aspect regarding overestimation of the computed wood volumes by the FGA and EGA is the assumption that the estimated volumes are supplied and exported to the outlet of the catchment, which may not be the case if the wood is being deposited along its way. The models do not consider the transfer of the wood along the river network (as for example the approaches by Franceschi et al., 2019 or Zischg et al., 2018)".*

*And also added some discussion after line 744 (original ms):*

*"The geospatial modelling results indicate areas of potential LW recruitment, however without precise information about the intensities occurring or the transfer and propagation through the river network."*

*And after line 742 (original ms):*

*"There is still a need to analyse and model the propagation of LW through the river network, by for example, applying hydraulic modelling (e.g., Ruiz-Villanueva et al., 2014) or the recently proposed network approaches as those applied to sediment transfer (Finch and Ruiz-Villanueva, 2022)."*

*Finch and Ruiz-Villanueva, 2022. Exploring the potential of the Graph Theory to large wood supply and transfer in river networks. EGU22-8232, https://doi.org/10.5194/egusphere-egu22-8232 EGU General Assembly 2022.*